# Episodic Multi-Task Learning with Heterogeneous Neural Processes

**Jiayi Shen**[1], **Xiantong Zhen**[1,2] [*], **Qi (Cheems) Wang**[3], **Marcel Worring**[1]

[1]University of Amsterdam, Netherlands, {j.shen, m.worring}@uva.nl

[2] Inception Institute of Artificial Intelligence, Abu Dhabi, UAE, zhenxt@gmail.com

[3] Kaiyuan Mathematical Sciences Institute, Changsha, China, hhq123go@gmail.com

## Abstract

This paper focuses on the data-insufficiency problem in multi-task learning within an episodic training setup. Specifically, we explore the potential of heterogeneous information across tasks and meta-knowledge among episodes to effectively tackle each task with limited data. Existing meta-learning methods often fail to take advantage of crucial heterogeneous information in a single episode, while multi-task learning models neglect reusing experience from earlier episodes. To address the problem of insufficient data, we develop Heterogeneous Neural Processes (HNPs) for the episodic multi-task setup. Within the framework of hierarchical Bayes, HNPs effectively capitalize on prior experiences as meta-knowledge and capture task-relatedness among heterogeneous tasks, mitigating data-insufficiency. Meanwhile, transformer-structured inference modules are designed to enable efficient inferences toward meta-knowledge and task-relatedness. In this way, HNPs can learn more powerful functional priors for adapting to novel heterogeneous tasks in each meta-test episode. Experimental results show the superior performance of the proposed HNPs over typical baselines, and ablation studies verify the effectiveness of the designed inference modules.

## 1 Introduction

Deep learning models have made remarkable progress with the help of the exponential increase in the amount of available training data [1]. However, many practical scenarios only have access to limited labeled data [2]. Such data-insufficiency sharply degrades the model's performance [2, 3]. Both meta-learning and multi-task learning have the potential to alleviate the data-insufficiency issue. Meta-learning can extract meta-knowledge from past episodes and thus enables rapid adaptation to new episodes with a few examples only [4–7]. Meanwhile, multi-task learning exploits the correlation among several tasks and results in more accurate learners for all tasks simultaneously [8–11]. However, the integration of meta-learning and multi-task learning in overcoming the data-insufficiency problem is rarely investigated.

In episodic training [4], existing meta-learning methods [4–7, 12, 13] in every meta-training or meta-test episode learn a single-task. In this paper, we refer to this conventional setting as *episodic single-task learning*. This setting restricts the potential for these models to explore task-relatedness within each episode, leaving the learning of multiple heterogeneous tasks in a single episode under-explored. We consider multiple tasks in each episode as *episodic multi-task learning*. The crux of episodic multi-task learning is to generalize the ability of exploring task-relatedness from meta-training to meta-test episodes. The differences between episodic single-task learning and episodic multi-task learning are illustrated in Figure 1. To be specific, we restrict the scope of the problem

---

[*]Currently with United Imaging Healthcare, Co., Ltd., China.

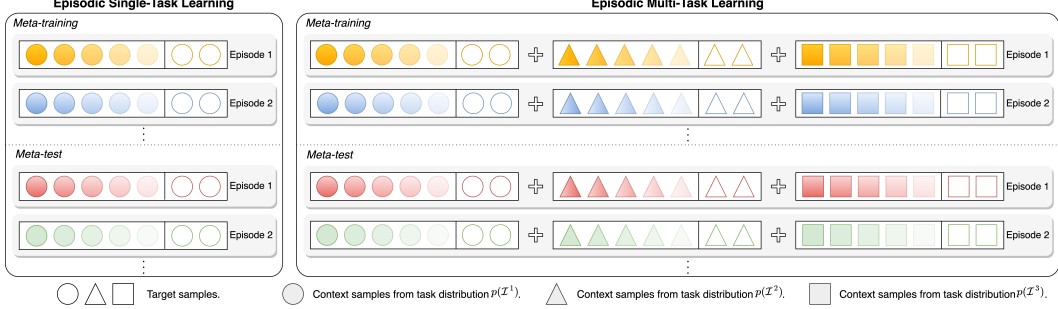

Figure 1: **Illustration of episodic multi-task learning.** Each row corresponds to a meta-training or meta-test episode. Different colors represent different label spaces among episodes; the same color with different shades represents different categories in the same task. Compared with episodic single-task learning, episodic multi-task learning simultaneously handles several related tasks in a single episode.

setup to the case where tasks in each meta-training or meta-test episode are heterogeneous but also relate to each other by sharing the same target space.

The neural process (NP) family [12, 13], as typical meta-learning probabilistic models [14], efficiently quantifies predictive uncertainty with limited data, making it in principle well-suited for tackling the problem of data-insufficiency. However, in practice, it is challenging for vanilla NPs [12] with a global latent variable to encode beneficial heterogeneous information in each episode. This issue is also known as the expressiveness bottleneck [15, 16], which weakens the model's capacity to handle insufficient data, especially when faced with diverse heterogeneous tasks.

To better resolve the data-insufficiency problem, we develop Heterogeneous Neural Processes (HNPs) for episodic multi-task learning. As a new member of the NP family, HNPs improve the expressiveness of vanilla NPs by introducing a hierarchical functional space with global and local latent variables. The remainder of this work is structured as follows: We introduce our method in Section (2). Related work is overviewed in Section (3). We report experimental results with analysis in Section (4), after which we conclude with a technical discussion, existing limitations, and future extensions. In detail, our technical contributions are two-fold:

- Built on the hierarchical Bayes framework, our developed HNPs can simultaneously generalize meta-knowledge from past episodes to new episodes and exploit task-relatedness across heterogeneous tasks in every single episode. This mechanism makes HNPs more powerful when encoding complex conditions into functional priors.
- We design transformer-structured inference modules to infer the hierarchical latent variables, capture task-relatedness, and learn a set of tokens as meta-knowledge. The designed modules can fuse the meta-knowledge and heterogeneous information from context samples in a unified manner, boosting the generalization capability of HNPs across tasks and episodes.

Experimental results show that the proposed HNPs together with transformer-structured inference modules, can exhibit superior performance on regression and classification tasks under the episodic multi-task setup.

## 2 Methodology

**Notations** [2]**.** We will now formally define episodic multi-task learning. For a single episode $\tau$, we consider $M$ heterogeneous but related tasks $\mathcal{I}_\tau^{1:M} = \{\mathcal{I}_\tau^m\}_{m=1}^M$. Notably, the subscript denotes an episode, while superscripts are used to distinguish tasks in this episode. In the episodic multi-task setup, tasks in a single episode are heterogeneous since they are sampled from different task distributions $\{p(\mathcal{I}^m)\}_{m=1}^M$, but are related at the same time as they share the target space $\mathcal{Y}_\tau$.

---

[2] For ease of presentation, we abbreviate a set $\{(\cdot)^m\}_{m=1}^M$ as $(\cdot)^{1:M}$, where $M$ is a positive integer. Likewise, $\{(\cdot)_o\}_{o=1}^O$ is abbreviated as $(\cdot)_{1:O}$. For convenience, the notation table is provided in Appendix B.

To clearly relate to the modeling of vanilla neural processes [12], this paper follows its nomenclature to define each task. Note that in vanilla neural processes *context* and *target* are often respectively called *support* and *query* in conventional meta-learning [4, 5]. Each task $\mathcal{I}_\tau^m$ contains a context set with limited training data $\mathcal{C}_\tau^m = \{\bar{x}_{\tau,i}^m, \bar{y}_{\tau,i}^m\}_{i=1}^{N_C}$ and a target set $\mathcal{T}_\tau^m = \{x_{\tau,j}^m, y_{\tau,j}^m\}_{j=1}^{N_T}$, where $N_C$ and $N_T$ are the numbers of context samples and target samples, respectively. $\bar{x}_{\tau,i}^m$ and $x_{\tau,j}^m$ represent features of context and target samples; while $\bar{y}_{\tau,i}^m, y_{\tau,j}^m \in \mathcal{Y}_\tau$ are their corresponding targets, where $i = 1, 2, ..., N_C; j = 1, 2, .., N_T; m = 1, 2, ..., M$. For simplicity, we denote the set of target samples and their corresponding ground-truths by $\mathbf{x}_\tau^m = \{x_{\tau,j}^m\}_{j=1}^{N_T}, \mathbf{y}_\tau^m = \{y_{\tau,j}^m\}_{j=1}^{N_T}$. For an episode $\tau$, episodic multi-task learning aims to perform simultaneously well on each corresponding target set $\mathcal{T}_\tau^m, m = 1, 2.., M$, given the collection of context sets $\mathcal{C}_\tau^{1:M}$.

For classification, this paper follows the protocol of meta models [4, 5, 17], such as $O$-way $K$-shot setup, clearly suffering from the data-insufficiency problem. Thus, episodic multi-task classification can be cast as a $M$-task $O$-way $K$-shot supervised learning problem. An episode has $M$ related classification tasks, and each of them has a context set with $K$ different instances from each of the $O$ classes [5]. It is worth mentioning that the target spaces of meta-training episodes do not overlap with any categories in those of meta-test episodes.

## 2.1 Modeling and Inference of Heterogeneous Neural Processes

We now present the proposed heterogeneous neural process. The proposed model inherits the advantages of multi-task learning and meta-learning, which can exploit task-relatedness among heterogeneous tasks and extract meta-knowledge from previous episodes. Next, we characterize the generative process, clarify the modeling within the hierarchical Bayes framework, and derive the approximate evidence lower bound (ELBO) in optimization.

**Generative Processes.** To get to our proposed method HNPs, we extend the distribution over a single function $p(f_\tau)$ as used in vanilla NPs to a joint distribution of multiple functions $p(f_\tau^{1:M})$ for all heterogeneous tasks in a single episode $\tau$. In detail, the underlying multi-task function distribution $p(f_\tau^{1:M})$ is inferred from a collection of context sets $\mathcal{C}_\tau^{1:M}$ and learnable meta-knowledge $\omega, \nu^{1:M}$. Note that $\omega$ represents the shared meta-knowledge for all tasks, and $\nu^m$ denotes the task-specific meta-knowledge corresponding to the task distribution $p(\mathcal{I}^m)$. Hence, we can formulate the predictive distribution for every single episode as follows:

$$p(\mathcal{T}_\tau^{1:M}|\mathcal{C}_\tau^{1:M}; \omega, \nu^{1:M}) = \int p(\mathbf{y}_\tau^{1:M}|\mathbf{x}_\tau^{1:M}, f_\tau^{1:M})p(f_\tau^{1:M}|\mathcal{C}_\tau^{1:M}; \omega, \nu^{1:M})df_\tau^{1:M}, \tag{1}$$

where $p(f_\tau^{1:M}|\mathcal{C}_\tau^{1:M}; \omega, \nu^m)$ denotes the data-dependent functional prior for multiple tasks of the episode $\tau$. The functional prior encodes context sets from all heterogeneous tasks and quantifies uncertainty in the functional space. Nevertheless, it is less optimal to characterize multi-task function generative processes with vanilla NPs, since the single latent variable limits the capacity of the latent space to specify the complicated functional priors. This expressiveness bottleneck in vanilla NPs is particularly severe for our episodic multi-task learning since each episode has diverse heterogeneous tasks with insufficient data.

**Modeling within the Hierarchical Bayes Framework.** To mitigate the expressiveness bottleneck of vanilla NPs, we model HNPs by parameterizing each task-specific function within a hierarchical Bayes framework. As illustrated in Figure 2, HNPs integrate a global latent representation $\mathbf{z}_\tau^m$ and a set of local latent parameters $\mathbf{w}_{\tau,1:O}^m$ to model each task-specific function $f_\tau^m$. Specifically, the latent variables are introduced at different levels: $\mathbf{z}_\tau^m$ encodes task-specific context information from $\mathcal{C}_\tau^m$ and $\nu^m$ in the representation level. $\mathbf{w}_{\tau,1:O}^m$ encode prediction-aware information for a task-specific decoder from $\mathcal{C}_\tau^{1:M}$ and $\omega$ in the parameter level, where $O$ is the dimension of the decoder. For example, the dimension is the size of the target space when performing classification tasks.

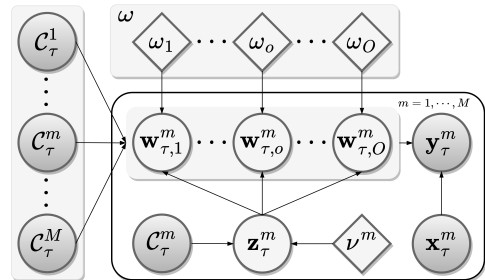

Figure 2: **Graphical model of the proposed HNPs in a single episode.** Filled shapes indicate observations. Probabilistic and deterministic variables are indicated by unfilled circles and diamonds, respectively.

Notably, each local latent parameter is conditioned on the global latent representation, which controls access to all context sets in the episode for the corresponding task. Our method differs from previous hierarchical architectures [16, 18–20] in the NP family since the local latent parameters of our HNPs are prediction-aware and explicitly constitute a decoder for the subsequent inference processes.

In practice, we assume that distributions of each task-specific function are conditionally independent. Thus, with the introduced hierarchical latent variables for each task in the episode, we can factorize the prior distribution over multiple functions in Eq. (1) into:

$$p(f_\tau^{1:M}|\mathcal{C}_\tau^{1:M};\omega,\nu^{1:M}) = \prod_{m=1}^{M} p(\mathbf{z}_\tau^m|\mathcal{C}_\tau^m;\nu^m)p(\mathbf{w}_{\tau,1:O}^m|\mathbf{z}_\tau^m,\mathcal{C}_\tau^{1:M};\omega), \tag{2}$$

where $p(\mathbf{z}_\tau^m|\mathcal{C}_\tau^m;\nu^m)$ and $p(\mathbf{w}_{\tau,1:O}^m|\mathbf{z}_\tau^m,\mathcal{C}_\tau^{1:M};\omega)$ are prior distributions of the global latent representation and the local latent parameters to induce the task-specific function distribution.

By integrating Eq. (2) into Eq. (1), we rewrite the modeling of HNPs in the following form:

$$p(\mathcal{T}_\tau^{1:M}|\mathcal{C}_\tau^{1:M};\omega,\nu^{1:M}) = \prod_{m=1}^{M} \int \Big\{ \int p(\mathbf{y}_\tau^m|\mathbf{x}_\tau^m,\mathbf{w}_{\tau,1:O}^m) \\ p(\mathbf{w}_{\tau,1:O}^m|\mathbf{z}_\tau^m,\mathcal{C}_\tau^{1:M};\omega)d\mathbf{w}_{\tau,1:O}^m \Big\}p(\mathbf{z}_\tau^m|\mathcal{C}_\tau^m;\nu^m)d\mathbf{z}_\tau^m, \tag{3}$$

where $p(\mathbf{y}_\tau^m|\mathbf{x}_\tau^m,\mathbf{w}_{\tau,1:O}^m)$ is the function distribution for the task $\mathcal{I}_\tau^m$ in HNPs. This distribution is obtained by the matrix multiplication of $\mathbf{x}_\tau^m$ and all local latent parameters $\mathbf{w}_{\tau,1:O}^m$.

Compared with most NP models [12, 16, 18, 19] employing only latent representations, HNPs infer both latent representations and parameters in the hierarchical architecture from multiple heterogeneous context sets and learnable meta-knowledge. Our model specifies a richer and more intricate functional space by leveraging the hierarchical uncertainty inherent in the context sets and meta-knowledge. This theoretically yields more powerful functional priors to induce multi-task function distributions.

Moreover, we claim that the developed model constitutes an *exchangeable stochastic process* and demonstrate this via Kolmogorov Extension Theorem [21]. Please refer to Appendix B for the proof.

**Approximate ELBO.** Since both exact functional posteriors and priors are intractable, we apply variational inference to the proposed HNPs in Eq. (3). This results in the approximate ELBO:

$$L_{\text{HNPs}}(\omega,\nu^{1:M},\theta,\phi) = \sum_{m=1}^{M} \Big\{ \mathbb{E}_{q_\theta(\mathbf{z}_\tau^m|\mathcal{T}_\tau^m;\nu^m)}\Big\{ \mathbb{E}_{q_\phi(\mathbf{w}_{\tau,1:O}^m|\mathbf{z}_\tau^m,\mathcal{T}_\tau^{1:M};\omega)}[\log p(\mathbf{y}_\tau^m|\mathbf{x}_\tau^m,\mathbf{w}_{\tau,1:O}^m)]$$

$$- \mathbb{D}_{\text{KL}}[q_\phi(\mathbf{w}_{\tau,1:O}^m|\mathbf{z}_\tau^m,\mathcal{T}_\tau^{1:M};\omega)||p_\phi(\mathbf{w}_{\tau,1:O}^m|\mathbf{z}_\tau^m,\mathcal{C}_\tau^{1:M};\omega)]\Big\} - \mathbb{D}_{\text{KL}}[q_\theta(\mathbf{z}_\tau^m|\mathcal{T}_\tau^m;\nu^m)||p_\theta(\mathbf{z}_\tau^m|\mathcal{C}_\tau^m;\nu^m)]\Big\}, \tag{4}$$

where $q_\theta(\mathbf{z}_\tau^m|\mathcal{T}_\tau^m;\nu^m)$ and $q_\phi(\mathbf{w}_{\tau,1:O}^m|\mathbf{z}_\tau^m,\mathcal{T}_\tau^{1:M};\omega)$ are variational posteriors of their corresponding latent variables. $\theta$ and $\phi$ are parameters of inference modules for $\mathbf{z}_\tau^m$ and $\mathbf{w}_{\tau,1:O}^m$, respectively. Following the protocol of vanilla NPs [12], the priors use the same inference modules as variational posteriors for tractable optimization. In this way, the KL-divergence terms in Eq. (4) encourage all latent variables inferred from the context sets to stay close to those inferred from the target sets, enabling effective function generation with few examples. Details on the derivation of the approximate ELBO and its tractable optimization are attached in Appendix C.

## 2.2 Transformer-Structured Inference Module

In order to infer the prior and variational posterior distributions in Eq. (4), it is essential to develop well-designed approximate inference modules. This is non-trivial and closely related to the performance of HNPs. Here we adopt a transformer structure as the inference module to better exploit task-relatedness from the meta-knowledge and the context sets in the episode. More specifically, the previously mentioned meta-knowledge $\omega = \omega_{1:O}$ and $\nu^{1:M}$ are instantiated as learnable tokens to induce the distributions of hierarchical latent variables in the proposed model.

Without loss of generality, in the next, we provide an example of transformer-structured inference modules for prior distributions in classification scenarios. Details of the inference modules in regression scenarios can be found in Appendix D. In Figure 3, a diagram of the transformer-structured

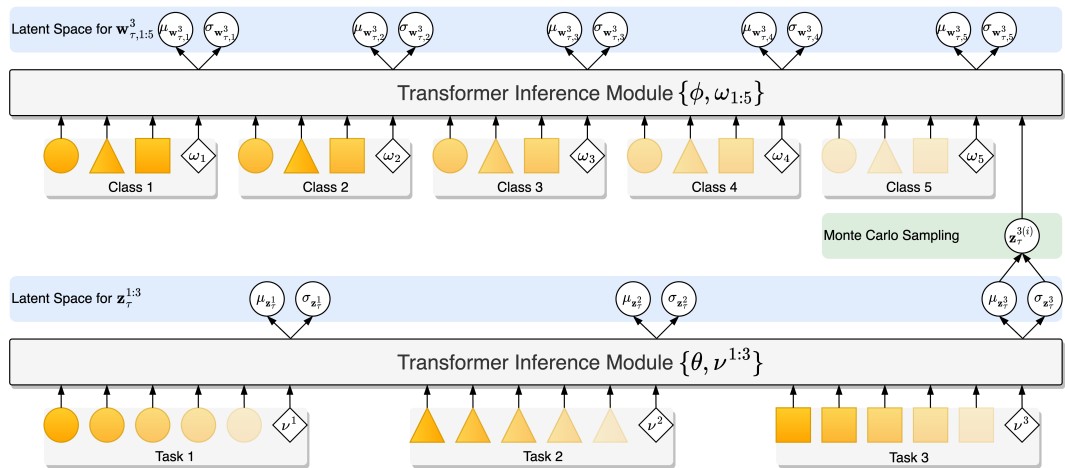

Figure 3: **A diagram of transformer-structured inference modules of HNPs for the first meta-training episode in Figure 1 under the `3-task 5-way 1-shot` setting.** For clarity, we display the inference process of the local latent parameters specific to the third task in the episode.

inference modules is displayed under the `3-task 5-way 1-shot` setting. In this case, the number of context samples is the same as the size of the target space, and thus we have $\mathcal{C}_\tau^m = \{\bar{x}_{\tau,o}^m, \bar{y}_{\tau,o}^m\}_{o=1}^O$, where $O$ is set as 5. In episodic training, labels in context sets are always available during inference.

**Transformer-Structured Inference Module $\{\theta, \nu^m\}$ for $\mathbf{z}_\tau^m$.** In the proposed HNPs, each global latent representation encodes task-specific information relevant to the considered task in the episode as $p_\theta(\mathbf{z}_\tau^m | \mathcal{C}_\tau^m; \nu^m)$. The learnable token $\nu^m$ preserves the meta-knowledge from previous episodes for specific tasks, which are sampled from the corresponding task distribution $p(\mathcal{I}^m)$. The role of $\nu^m$ is to help the model adapt efficiently to such specific tasks in meta-test episodes.

In detail, we set the dimension of the learnable token $\nu^m$ to the same as that of the features $\bar{x}_{\tau,1:O}^m$. Then the transformer-structured inference module $\theta$ fuses them in a unified manner by taking $[\bar{x}_{\tau,1:O}^m; \nu^m]$ as the input. The module $\theta$ outputs the mean and variance of the corresponding prior distribution. The inference steps for the global latent representation $\mathbf{z}_\tau^m$ are:

$$[\widetilde{x}_{\tau,1:O}^m; \widetilde{\nu}^m] = \texttt{MSA}(\texttt{LN}([\bar{x}_{\tau,1:O}^m; \nu^m])) + [\bar{x}_{\tau,1:O}^m; \nu^m], \tag{5}$$

$$[\widehat{x}_{\tau,1:O}^m; \widehat{\nu}^m] = \texttt{MLP}(\texttt{LN}([\widetilde{x}_{\tau,1:O}^m; \widetilde{\nu}^m])) + [\widetilde{x}_{\tau,1:O}^m; \widetilde{\nu}^m], \tag{6}$$

$$p_\theta(\mathbf{z}_\tau^m | \mathcal{C}_\tau^m; \nu^m) = \mathcal{N}(\mathbf{z}_\tau^m; \mu_{\mathbf{z}_\tau^m}, \sigma_{\mathbf{z}_\tau^m}), \tag{7}$$

where $\mu_{\mathbf{z}_\tau^m} = \texttt{MLP}(\widehat{\nu}^m), \sigma_{\mathbf{z}_\tau^m} = \texttt{Softplus}(\texttt{MLP}(\widehat{\nu}^m))$. The transformer-structured inference module includes a multi-headed self-attention (`MSA`) and three multi-layer perceptrons (`MLP`). The layer normalization (`LN`) is "pre-norm" as done in [22]. `Softplus` is the activation function to output the appropriate value as the variance of the prior distribution [23].

**Transformer-Structured Inference Module $\{\phi, \omega_{1:O}\}$ for $\mathbf{w}_{\tau,1:O}^m$.** Likewise, each learnable token $\omega_o$ corresponds to a local latent parameter $\mathbf{w}_{\tau,o}^m$. With the learnable tokens $\omega_{1:O}$, we reformulate the prior distribution of local latent parameters as $p_\phi(\mathbf{w}_{\tau,1:O}^m | \mathbf{z}_\tau^m, \mathcal{C}_\tau^{1:M}; \omega_{1:O})$. In this way, we learn the shared knowledge, inductive biases across all tasks, and their distribution at a parameter level, which in practical settings can capture epistemic uncertainty.

To be specific, the prior distribution can be factorized as $\prod_{o=1}^O p_\phi(\mathbf{w}_{\tau,o}^m | \mathbf{z}_\tau^m, \mathcal{C}_\tau^{1:M}; \omega_o)$, where all local latent parameters are assumed to be conditionally independent. For each local latent parameter $\mathbf{w}_{\tau,o}^m$, the transformer-structured inference module $\phi$ takes $[\bar{x}_{\tau,o}^{1:M}, \omega_o]$ as input and outputs the corresponding prior distribution, where $\bar{x}_{\tau,o}^{1:M}$ are deep features from the same class $o$ in the episode and $\omega_o$ is the corresponding learnable token. Here the inference steps for $\mathbf{w}_{\tau,o}^m$ are as follows:

$$[\widetilde{x}_{\tau,o}^{1:M}; \widetilde{\omega}_o] = \texttt{MSA}(\texttt{LN}([\bar{x}_{\tau,o}^{1:M}; \omega_o])) + [\bar{x}_{\tau,o}^{1:M}; \omega_o], \tag{8}$$

$$[\widehat{x}_{\tau,o}^{1:M}; \widehat{\omega}_o] = \texttt{MLP}(\texttt{LN}([\widetilde{x}_{\tau,o}^{1:M}; \widetilde{\omega}_o])) + [\widetilde{x}_{\tau,o}^{1:M}; \widetilde{\omega}_o], \tag{9}$$

$$p_\phi(\mathbf{w}_{\tau,o}^m | \mathbf{z}_\tau^m, \mathcal{C}_\tau^{1:M}; \omega_o) = \mathcal{N}(\mathbf{w}_{\tau,o}^m; \mu_{\mathbf{w}_{\tau,o}^m}, \sigma_{\mathbf{w}_{\tau,o}^m}), \tag{10}$$

where $\mu_{\mathbf{w}_{\tau,o}^m} = \texttt{MLP}(\widehat{\omega}_o, \mathbf{z}_\tau^{m(i)})$, $\sigma_{\mathbf{w}_{\tau,o}^m} = \texttt{Softplus}(\texttt{MLP}(\widehat{\omega}_o, \mathbf{z}_\tau^{m(i)}))$. $\mathbf{z}_\tau^{m(i)}$ is a Monte Carlo sample from the variational posterior of the corresponding global latent representation during meta-training.

Both transformer-structured inference modules use the refined tokens $\widehat{\nu}^m$ and $\widehat{\omega}_o$ to obtain a global latent representation and a local latent parameter, respectively. The introduced tokens preserve the specific meta-knowledge for each latent variable during inference. Compared with the $\theta$-parameterised inference module exploring the intra-task relationships, the $\phi$-parameterised inference module enables the exploitation of the inter-task relationships to reason over each local latent parameter. Thus, the introduced tokens can be refined with relevant information from the heterogeneous context sets. By integrating meta-knowledge and heterogeneous context sets, HNPs can reduce the negative transfer of task-specific knowledge among heterogeneous tasks in each episode. Please refer to Appendix E for algorithms.

## 3 Related Work

**Multi-Task Learning.** Multi-task learning can operate in various settings [9]. Here we roughly separate the settings of MTL into two branches: (1) Single-input multi-output (SIMO) [24–30], where tasks are defined by different supervision information for the same input. (2) Multi-input multi-output (MIMO) [11, 10, 31–34], where heterogeneous tasks follow different data distributions. This work considers the MIMO setup of multi-task learning with episodic training.

In terms of modeling methods, from a processing perspective, existing MTL methods can be roughly categorized into two groups: (1) Probabilistic MTL methods [11, 19, 35–41], which employ the Bayes framework to characterize probabilistic dependencies among tasks. (2) Deep MTL models [10, 32, 24–26, 42–48], which directly utilize deep neural networks to discover information-sharing mechanisms across tasks. However, deep MTL models rely on large amounts of training data and tend to overfit when encountering the data-insufficiency problem. Meanwhile, previous probabilistic MTL methods consider a small number of tasks that occur at the same time, limiting their applicability in real-world systems.

**Meta-Learning.** Meta-learning aims to find strategies to quickly adapt to unseen tasks with a few examples [49, 4, 5, 50]. There exist a couple of branches in meta-learning methods, such as metrics-based methods [6, 51–57] and optimization-based methods [5, 58–68]. Our paper focuses on a probabilistic meta-learning method, namely neural processes, that can quantify predictive uncertainty. Models in this family [7, 12, 13, 15, 16, 18, 69–73] can approximate stochastic processes in neural networks. Vanilla NPs [12] usually encounter the expressiveness bottleneck because their functional priors are not rich enough to generate complicated functions [15, 16]. [7] introduces deterministic variables to model predictive distributions for meta-learning scenarios directly. Most NP-based methods only focus on a single task during inference [7, 12, 15, 16, 14], which leaves task-relatedness between heterogeneous tasks in a single episode an open problem.

This paper combines multi-task learning and meta-learning paradigms to tackle the data-insufficiency problem. Our work shares the high-level goal of exploiting task-relatedness in an episode with [19, 74, 75]. Concerning the multi-task scenarios, the main differences are: [19, 74, 75] handles multiple attributes and multi-sensor data under the SIMO setting, while our work performs for the MIMO setting where tasks are heterogeneous and distribution shifts exist. Moreover, [76] theoretically addresses the conclusion that MTL methods are powerful and efficient alternatives to gradient-based meta-learning algorithms. However, our method inherits the advantages of multi-task learning and meta-learning: simultaneously generalizing meta-knowledge from past to new episodes and exploiting task-relatedness across heterogeneous tasks in every single episode. Thus, our method is more suitable for solving the data-insufficiency problem. Intuitive comparisons with related paradigms such as *cross-domain few-shot learning* [77–82], *multimodal meta-learning* [83–87, 56] and *cross-modality few-shot learning* [88–90] are provided in Appendix A.

# 4 Experiments

We evaluate the proposed HNPs and baselines on three benchmark datasets under the episodic multi-task setup. Sec. 4.1 and Sec. 4.2 provide experimental results for regression and classification, respectively. Ablation studies are in Sec. 4.3. More comparisons with recent works on extra datasets are provided in Appendix F. Additional results under the convectional MIMO setup without episodic training can be found in Appendix G & H.

## 4.1 Episodic Multi-Task Regression

**Dataset and Settings.** To evaluate the benefit of HNPs over typical NP baselines in uncertainty quantification, we conduct experiments in several 1D regression tasks. The baselines include conditional neural processes (CNPs [13]), vanilla neural processes (NPs [12]), and attentive neural processes (ANPs [15]). As a toy example, we construct multiple tasks with different task distributions: each task's input set is defined on separate intervals without overlap.

Given four different tasks in an episode, their input sets are $\mathbf{x}_\tau^{1:4}$. Each input set contains a few instances, drawn uniformly at random from separate intervals, such as $\mathbf{x}_\tau^1 \in [-4, -2)$, $\mathbf{x}_\tau^2 \in [-2, 0)$, $\mathbf{x}_\tau^3 \in [0, 2)$, and $\mathbf{x}_\tau^4 \in [2, 4)$. All tasks in an episode are related by sharing the same ground truth function. Following [12, 18], function-fitting tasks are generated with Gaussian processes (GPs). Here a zero mean Gaussian process $y^{(0)} \sim \mathcal{GP}(0, k(\cdot, \cdot))$ is used to produce $\mathbf{y}_\tau^{1:4}$ for the inputs from all tasks $\mathbf{x}_\tau^{1:4}$. A radial basis kernel $k(x, x') = \sigma^2 \exp(-(x - x')^2)/2l^2)$, with $l = 0.4$ and $\sigma = 1.0$ is used.

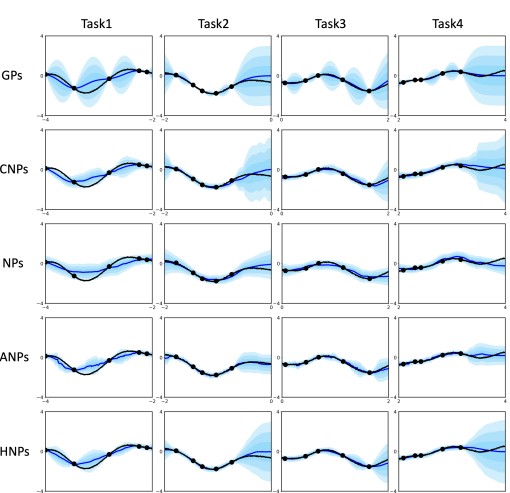

Figure 4: **Performance comparisons on the episodic multi-task** 1**-D function regression using** 5 **context points (black dots) for each task.** Black curves are ground truth, and blue ones are predicted results. The shadow regions are ±3 standard derivations from the mean [18].

**Results and Discussions.** As shown in Figure 4, CNPs, ANPs, and our HNPs exhibit more reasonable uncertainty than NPs in Figure 4: lower variances are predicted around observed (context) points with higher variances around unobserved points. Furthermore, NPs and ANPs detrimentally impact the smoothness of the predicted curves, whereas HNPs yield smoother predictive curves with reliable uncertainty estimation. These observations suggest that integrating correlation information across related tasks and meta-knowledge in HNPs can improve uncertainty quantification in multi-task regression.

To quantify uncertainty we use the average negative log-likelihood (the lower, the better). As shown in Table 1, our HNPs achieve a lower average negative log-likelihood than baselines, demonstrating our method's effectiveness in uncertainty estimation.

Table 1: **Average negative log-likelihoods over target points from all tasks.**

| Methods | CNPs | NPs | ANPs | HNPs |
|---------|------|-----|------|------|
| Avg. NLL | 0.0935 | 0.8649 | -0.1165 | -0.5207 |

## 4.2 Episodic Multi-task Classification

**Datasets and Settings.** We use `Office-Home` [91] and `DomainNet` [92] as episodic multi-task classification datasets. `Office-Home` contains images from four domains: Artistic (A), Clipart (C), Product (P) and Real-world (R). Each domain contains images from 65 categories collected from office and home environments. Note that all domains share the whole target space. The numbers of meta-training classes and meta-test classes are 40 and 25, respectively. There are about 15, 500 images in total. `DomainNet` has six distinct domains: Clipart, Infograph, Painting, Quickdraw, Real and Sketch. It includes approximately 0.6 million images distributed over 345 categories. The

Table 2: **Comparative results (95% confidence interval) for episodic multi-task classification on** `Office-Home` **and** `DomainNet`**.** Best results are indicated in bold.

| | Office-Home | | | | DomainNet | | | |
| | 4-task 5-way | | 4-task 20-way | | 6-task 5-way | | 6-task 20-way | |
| Method | 1-shot | 5-shot | 1-shot | 5-shot | 1-shot | 5-shot | 1-shot | 5-shot |
|---|---|---|---|---|---|---|---|---|
| ERM [93] | 66.04 $\pm$0.61 | 73.62 $\pm$0.55 | 39.25 $\pm$0.24 | 47.14 $\pm$0.18 | 59.95 $\pm$0.52 | 68.52 $\pm$0.44 | 38.62 $\pm$0.22 | 47.85 $\pm$0.20 |
| VMTL [11] | 49.71 $\pm$0.48 | 65.75 $\pm$0.47 | 27.50 $\pm$0.14 | 42.82 $\pm$0.13 | 42.24 $\pm$0.39 | 57.37 $\pm$0.43 | 18.05 $\pm$0.11 | 31.38 $\pm$0.15 |
| MAML [5] | 60.58 $\pm$0.60 | 75.29 $\pm$0.53 | 34.29 $\pm$0.19 | 48.39 $\pm$0.20 | 53.21 $\pm$0.46 | 65.24 $\pm$0.47 | 17.10 $\pm$0.12 | 20.35 $\pm$0.14 |
| Proto. Net. [6] | 57.19 $\pm$0.53 | 74.97 $\pm$0.46 | 32.72 $\pm$0.18 | 49.75 $\pm$0.16 | 53.71 $\pm$0.48 | 68.80 $\pm$0.42 | 31.90 $\pm$0.19 | 47.59 $\pm$0.18 |
| DGPs [94] | 65.89 $\pm$0.53 | 79.96 $\pm$0.38 | 31.48 $\pm$0.18 | 49.46 $\pm$0.18 | 50.93 $\pm$0.42 | 63.32 $\pm$0.38 | 25.46 $\pm$0.15 | 38.63 $\pm$0.17 |
| CNPs [13] | 43.33 $\pm$0.56 | 55.07 $\pm$0.63 | 10.57 $\pm$0.10 | 12.02 $\pm$0.11 | 37.90 $\pm$0.45 | 40.53 $\pm$0.44 | 5.12 $\pm$0.10 | 5.14 $\pm$0.10 |
| NPs [12] | 33.66 $\pm$0.48 | 53.99 $\pm$0.60 | 5.25 $\pm$0.16 | 11.40 $\pm$0.11 | 20.58 $\pm$0.51 | 20.53 $\pm$0.53 | 5.12 $\pm$0.09 | 5.11 $\pm$0.09 |
| TNP-D [95] | 65.49 $\pm$0.53 | 78.94 $\pm$0.43 | 41.61 $\pm$0.22 | 59.19 $\pm$0.21 | 49.10 $\pm$0.42 | 67.39 $\pm$0.40 | 28.83 $\pm$0.17 | 47.69 $\pm$0.18 |
| HNPs | **76.29** $\pm$0.51 | **80.80** $\pm$0.42 | **51.82** $\pm$0.23 | **59.97** $\pm$0.18 | **62.36** $\pm$0.53 | **69.38** $\pm$0.42 | **39.32** $\pm$0.23 | **48.56** $\pm$0.19 |

numbers of meta-training classes and meta-test classes are 276 and 69, respectively. Here one domain corresponds to a specific task distribution in the episodic multi-task setting.

When it comes to the episodic multi-task classification, we compare HNPs with the following three branches: (1) *Multi-task learning methods*: ERM [93] directly expands the training set of the current task with samples of related tasks. VMTL [11] is one of the state-of-the-art under the MIMO setting of multi-task learning. (2) *Meta-learning methods*: MAML [5], Proto.Net [6] and DGPs [94] address each task separately with no mechanism to leverage task-relatedness in a single episode. (3) *Methods from the NP family*: CNPs [13] and NPs [12] are established methods in the NP family. TNP-D [95] is recent NP work in sequential decision-making for a single task in each episode.

**Results and Discussions.** The experimental results for episodic multi-task classification on `Office-Home` and `DomainNet` are reported in Table 2. We use the average accuracy across all task distributions as the evaluation metric. It can be seen that HNPs consistently outperform all baseline methods, demonstrating the effectiveness of HNPs in handling each task with limited data under the episodic multi-task classification setup.

NPs and CNPs do not work well under all episodic multi-task classification cases. This can be attributed to their limited expressiveness of the global representation and the weak capability to extract discriminative information from multiple contexts. In contrast, HNPs explicitly abstract discriminative information for each task in the episode with the help of local latent parameters, enhancing the expressiveness of the functional prior.

We also find that HNPs significantly surpass other baselines on `1-shot` `Office-Home` and `DomainNet`, both under the `4/6-task 5-way` and `4/6-task 20-way` settings. This further implies that HNPs can circumvent the effect of the problem of data-insufficiency by simultaneously exploiting task-relatedness across heterogeneous tasks and meta-knowledge among episodes.

## 4.3 Ablation Studies

**Influence of Hierarchical Latent Variables.** We first investigate the roles of the global latent representation $\mathbf{z}_\tau^m$ and the local latent parameters $\mathbf{w}_{\tau,1:O}^m$ by leaving out individual inference modules. These experiments are performed on `Office-home` under the `4-task 5-way 1-shot` setting. We report the detailed performance for tasks sampled from a single task distribution (A/C/P/R) and the average accuracy across all task distributions (Avg.) in Table 3. The variants without specific latent variables are included in the comparison by removing the corresponding inference modules.

As shown in Table 3, both $\mathbf{z}_\tau^m$ and $\mathbf{w}_{\tau,1:O}^m$ benefit overall performance. Our method with hierarchical latent variables performs 9.20% better than the variant without both latent variables, 3.00% better than the variant without $\mathbf{z}_\tau^m$, and 5.18% better than the variant without $\mathbf{w}_{\tau,1:O}^m$. This indicates that latent variables of HNPs complement each other in representing con-

Table 3: **Effectiveness of global latent representations** $\mathbf{z}_\tau^m$ **and local latent parameters** $\mathbf{w}_{\tau,1:O}^m$ **in the model.** ✓ and ✗ denote whether the variants of HNPs have the corresponding latent variable or not.

| $\mathbf{z}_\tau^m$ | $\mathbf{w}_{\tau,1:O}^m$ | A | C | P | R | Avg. |
|---|---|---|---|---|---|---|
| ✗ | ✗ | 62.64 $\pm$0.72 | 56.87 $\pm$0.71 | 75.18 $\pm$0.79 | 73.68 $\pm$0.77 | 67.09 $\pm$0.63 |
| ✗ | ✓ | 69.39 $\pm$0.60 | 63.10 $\pm$0.61 | 80.66 $\pm$0.67 | 79.99 $\pm$0.62 | 73.29 $\pm$0.51 |
| ✓ | ✗ | 67.02 $\pm$0.67 | 60.70 $\pm$0.69 | 78.26 $\pm$0.76 | 78.47 $\pm$0.72 | 71.11 $\pm$0.59 |
| ✓ | ✓ | **73.31** $\pm$0.63 | **64.92** $\pm$0.68 | **83.38** $\pm$0.66 | **83.54** $\pm$0.64 | **76.29** $\pm$0.51 |

text sets from multiple tasks and meta-knowledge. The variant without $\mathbf{w}_{\tau,1:O}^m$ underperforms the variant without $\mathbf{z}_\tau^m$ by 2.18%, in terms of the average accuracy. This demonstrates that $\mathbf{z}_\tau^m$ suffers more from the expressiveness bottleneck than $\mathbf{w}_{\tau,1:O}^m$, weakening the models' discriminative ability. For classification, local latent parameters are more crucial than a global latent representation in revealing the discriminating knowledge from multiple heterogeneous context sets.

**Influence of Transformer-Structured Inference Modules.**   To further understand our transformer-structured inference modules (Trans. w learnable tokens), we examine the performance against two other options: inference modules that solely utilize a multi-layer perceptron (MLP) and the variants that do not incorporate any learnable tokens (Trans. w/o learnable tokens). We also compare the probabilistic and deterministic versions of such inference modules. The deterministic variants consider the deterministic embedding for the hierarchical latent variables.

As shown in Table 4, our inference modules consistently outperform the variants, regardless of whether the inference network is probabilistic or deterministic. When using the probabilistic one, our inference modules respectively achieve 1.04% and 2.99% performance gains over Trans. w/o learnable tokens and MLP under the `4-task 5-way 1-shot` setting. This implies the importance of learn-

Table 4: **Performance comparisons between our transformer inference modules (Trans. w learnable tokens) and other alternatives.**

| Inference networks | | 1-shot | 5-shot |
|---|---|---|---|
| | MLP | 64.93 $_{\pm0.66}$ | 72.39 $_{\pm0.56}$ |
| Deterministic | Trans. w/o learnable tokens | 70.22 $_{\pm0.62}$ | 76.15 $_{\pm0.54}$ |
| | Trans. w learnable tokens | 70.61 $_{\pm0.56}$ | 76.70 $_{\pm0.50}$ |
| | MLP | 73.30 $_{\pm0.59}$ | 77.94 $_{\pm0.48}$ |
| Probabilistic | Trans. w/o learnable tokens | 75.25 $_{\pm0.55}$ | 80.42 $_{\pm0.47}$ |
| | Trans. w learnable tokens | **76.29** $_{\pm\mathbf{0.51}}$ | **80.80** $_{\pm\mathbf{0.42}}$ |

able tokens and task-relatedness in formulating transformer-structured inference modules, which reduces negative transfer among heterogeneous tasks in each meta-test episode. Moreover, the variants with probabilistic inference modules consistently beat deterministic ones in performance, demonstrating the advantages of considering uncertainty during modeling and inference.

**Effects of Different Ways to Generate Local Latent Parameters.**   We investigate the effects of different ways to generate each $\mathbf{w}_{\tau,o}^m$ from the shared condition $\mathbf{z}_\tau^m$ and $\mathcal{C}_\tau^{1:M}$. Given a Monte Carlo sample of global latent variables as $\mathbf{z}_\tau^{m(i)}$, in Table 5, we compare with two alternatives:

Table 5: **Performance comparisons of different implementations of generating each local latent parameter $\mathbf{w}_{\tau,o}^m$ from the condition $\mathbf{z}_\tau^m$ and $\mathcal{C}_\tau^{1:M}$.**

| Methods | A | C | P | R | Avg. |
|---|---|---|---|---|---|
| Concat | 65.69 $_{\pm0.59}$ | 58.64 $_{\pm0.61}$ | 77.54 $_{\pm0.68}$ | 77.10 $_{\pm0.64}$ | 69.74 $_{\pm0.51}$ |
| Add | 69.92 $_{\pm0.69}$ | 63.73 $_{\pm0.71}$ | 78.81 $_{\pm0.77}$ | 79.03 $_{\pm0.78}$ | 72.87 $_{\pm0.61}$ |
| Ours | **73.31** $_{\pm\mathbf{0.63}}$ | **64.92** $_{\pm\mathbf{0.68}}$ | **83.38** $_{\pm\mathbf{0.66}}$ | **83.54** $_{\pm\mathbf{0.64}}$ | **76.29** $_{\pm\mathbf{0.51}}$ |

1) `Concat` directly concatenates each context feature and $\mathbf{z}_\tau^{m(i)}$, and takes the concatenation as inputs of the transformer-structured inference network $\phi$. 2) `Add` sums up each context feature and $\mathbf{z}_\tau^{m(i)}$ and takes the result as the input. 3) `Ours` incorporates $\mathbf{z}_\tau^{m(i)}$ into the transformer-structured inference module by merging it with the refined learnable tokens in Eq. (10). As shown in Table 5, `Ours` consistently performs the best. This implies that incorporating the conditional variables into the inference module is more effective than the direct combinations of $\mathbf{z}_\tau^{m(i)}$ and instance features.

**Effects of More "Shots" or "Classes".**   To investigate the effects of more "shots" or "classes" in the episodic multi-task classification setup, we conduct experiments by increasing $K$ or $O$ in the defined $M$-`task` $O$-`way` $K$-`shot` setup.

As shown in Table 6, the proposed HNPs have more advantages over the baseline method with the context data points below ten shots. With shots larger than ten, both methods will reach a performance bottleneck.

Table 6: **Performance comparisons on `Office-Home` under the `4-task 5-way K-shot` setup.**

| Methods | 1-shot | 5-shot | 10-shot | 20-shot |
|---|---|---|---|---|
| TNP-D | 65.49 $_{\pm0.53}$ | 78.94 $_{\pm0.43}$ | 80.81 $_{\pm0.32}$ | 81.12 $_{\pm0.68}$ |
| HNPs | 76.29 $_{\pm0.51}$ | 80.80 $_{\pm0.42}$ | 81.28 $_{\pm0.38}$ | 81.56 $_{\pm0.36}$ |

Moreover, Table 7 shows that our method consistently outperforms the baseline method as the number of classes increases from 20 to 40 in step 5. However, the performance gap between them narrows slightly with more classes. The main reason could be that the setting with more classes suffers from less data insufficiency.

Table 7: **Performance comparisons on** `DomainNet` **under the** `6-task O-way 1-shot` **setup.**

| Methods | 5-way | 20-way | 25-way | 30-way | 35-way | 40-way |
|---|---|---|---|---|---|---|
| TNP-D | 49.10 $\pm_{0.42}$ | 28.83 $\pm_{0.17}$ | 25.93 $\pm_{0.14}$ | 24.08 $\pm_{0.12}$ | 22.62 $\pm_{0.11}$ | 21.64 $\pm_{0.53}$ |
| HNPs | 62.36 $\pm_{0.53}$ | 39.32 $\pm_{0.23}$ | 35.72 $\pm_{0.19}$ | 32.27 $\pm_{0.17}$ | 31.27 $\pm_{0.14}$ | 29.31 $\pm_{0.13}$ |

**Sensitivity to the Number of Monte Carlo Samples.** For the hierarchical latent variables in the HNPs, we investigate the model's sensitivity to the number of Monte Carlo samples. Specifically, the sampling number of the global latent representation $\mathbf{z}_\tau^m$ and local latent parameters $\mathbf{w}_{\tau,1:O}^m$ varies from 1 to 30. We examine on `Office-Home` under the `4-task 5-way 1-shot` setting. In Figure 5, the runtime per iteration grows rapidly as the number of samples increases. However, there is no clear correlation between the performance and the num-

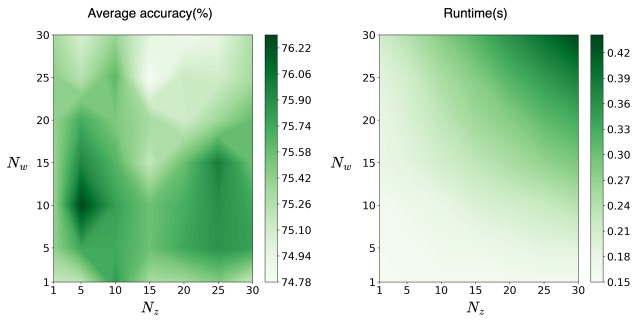

Figure 5: **Average accuracy and runtime of HNPs with different numbers of Monte Carlo samples.** $N_z$ and $N_w$ are sampling numbers of $\mathbf{z}_\tau^m$ and $\mathbf{w}_{\tau,1:O}^m$, respectively.

ber of Monte Carlo samples. There are two sweet spots in terms of average accuracy, one of which has favorable computation time. Hence, we set $N_z$ and $N_w$ to 5 and 10, respectively.

We also investigate the inference time of NP-based models per iteration on `Office-Home` under the `4task5way1shot` setup. As shown in Table 8, our model needs more inference time than other NP-based meth-

Table 8: **Inference time of different NP-based methods.**

| Methods | CNPs | NPs | TNP-D | HNPs |
|---|---|---|---|---|
| Inference time(s) | 0.04 | 0.05 | 0.08 | 0.15 |

ods for performance gains. The cost mainly comes from inferring the designed hierarchical latent variables; however, we consider this a worthwhile trade-off for the extra performance.

## 5 Conclusion

**Technical Discussion.** This work develops heterogeneous neural processes by introducing hierarchical latent variables and transformer-structured inference modules for episodic multi-task learning. With the help of heterogeneous context information and meta-knowledge, the proposed model can exploit task-relatedness, reason about predictive function distributions, and efficiently distill past knowledge to unseen heterogeneous tasks with limited data.

**Limitation & Extension.** Although the hierarchical probabilistic framework could mitigate the expressiveness bottleneck, the model needs more inference time than other NP-based methods for performance gains. Besides, the proposed method requires the target space to be the same across all tasks in a single episode. This requirement could limit the method's applicability in realistic scenarios where target spaces may differ across tasks. Our work could be extended to the new case without the shared target spaces, where the model should construct higher-order task-relatedness to improve knowledge sharing among tasks. Our code [3] is provided to facilitate such extensions.

## Acknowledgment

This work is financially supported by the Inception Institute of Artificial Intelligence, the University of Amsterdam and the allowance Top consortia for Knowledge and Innovation (TKIs) from the Netherlands Ministry of Economic Affairs and Climate Policy.

---

[3] https://github.com/autumn9999/HNPs.git

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
