# Appendix for "Episodic Multi-Task Learning with Heterogeneous Neural Processes"

**Jiayi Shen**[1], **Xiantong Zhen**[1,2], **Qi (Cheems) Wang**[3], **Marcel Worring**[1]
[1]University of Amsterdam, Netherlands, {j.shen, m.worring}@uva.nl
[2] Inception Institute of Artificial Intelligence, Abu Dhabi, UAE, zhenxt@gmail.com
[3] Kaiyuan Mathematical Sciences Institute, Changsha, China, hhq123go@gmail.com

## Contents

37th Conference on Neural Information Processing Systems (NeurIPS 2023).

# A    Frequently Asked Questions

In this section, we list frequently asked questions from researchers who help proofread this manuscript. These raised questions might also be relevant for others and help in better understanding the paper, so we include more detailed discussions here.

**Connections between different settings.**    This work considers the multi-input multi-output setting of multi-task learning under the episodic training mechanism.

As shown in Table 1, we use "Heterogeneous tasks" to distinguish the different branches of multi-task learning: (1) *single-input multi-output* (SIMO) considers different tasks which have the same input and different supervision information. (2) *multi-input multi-output* (MIMO) considers heterogeneous tasks, which have different inputs and follow different data distributions. All tasks are related since they share the target space. This setting encourages deep models to deal with the insufficient data of each task by aggregating the training data from related tasks in the spirit of data augmentation.

Meanwhile, "Episodic training" is used to describe the data-feeding strategy. *Multi-task meta-learning* also benefits from episodic training, but it follows the SIMO setting in every single episode and cannot sufficiently handle heterogeneous tasks. In our work, *episodic multi-task learning* is designed based on the MIMO setting, suffering from distribution shifts between heterogeneous tasks. In addition, we note that *conventional meta-learning* follows the "Episodic training" mechanism but focuses on single-task learning in each episode. Thus, "Heterogeneous tasks" is not available here (-). More details are left in Table (1).

Table 1: **Connections between different settings.** The symbols ✓ and ✗ indicate whether or not the specific setting has the corresponding characteristic.

| Settings | Methods | Episodic training | Heterogeneous tasks |
|---|---|---|---|
| *single-input multi-output* (SIMO) | [1–7] | ✗ | ✗ |
| *multi-input multi-output* (MIMO) | [8–13] | ✗ | ✓ |
| *conventional meta-learning* | [14–19] | ✓ | - |
| *multi-task meta-learning* | [20–26] | ✓ | ✗ |
| *episodic multi-task learning* | This paper | ✓ | ✓ |

**Problem scope.**    In episodic multi-task learning, we restrict the scope of the problem to the case where tasks in the same episode are related and share the same target space. There are two main reasons: (1) we follow the MIMO setting of multi-task learning in every single episode, where the same target space assures the existence of the knowledge shared among tasks. (2) As demonstrated in recent works [27, 28], meta-learning tasks generated from the same categories or taxonomic clusters are closer. This also implies that tasks with the same target space are related.

**Differences from other episodic single-task setups.**    Based on episodic training, there are several approaches related to the setting of episodic multi-task learning: (1) *cross-domain few-shot learning* addresses few-shot learning under domain shifts [29]. Several models [30, 29, 31–34] train a model on a single source domain or several source domains and then generalize it to other domains. In contrast, our research emphasizes the domain shifts within individual episodes rather than among them. (2) *multimodal meta-learning* extends few-shot learning from a single input-label domain to multiple different input-label domains [35]. These methods [36, 35, 37–40] design a meta-learner that can handle tasks from distinct distributions in sequence. Our work centers on simultaneously dealing with several related tasks within a meta-training or meta-test episode. (3) *cross-modality few-shot learning* [41–43] leverages semantic information (e.g., word vectors) to augment the performance of visual tasks and not among visual tasks only. The aforementioned approaches exclusively address single-task learning per episode, while our work concurrently tackles multiple heterogeneous and related tasks within each meta-training or meta-test episode. Intuitive comparisons with the approaches are shown in Figure 1.

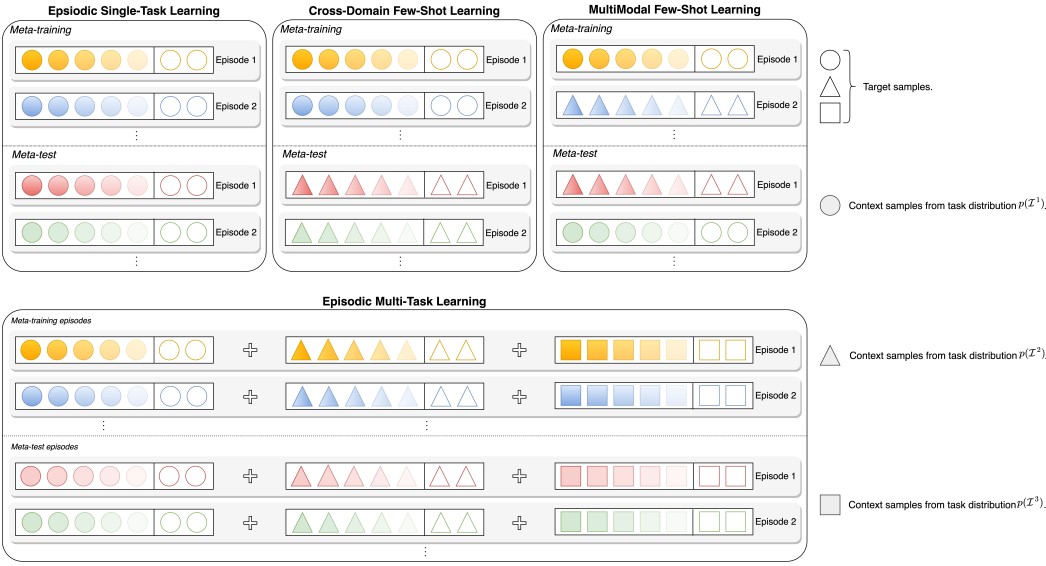

Figure 1: **Differences from other episodic single-task setups**. Each row corresponds to an episode. Different color denotes different categories; the same color with different shades represents different categories in the same task. Episodic multi-task learning is orthogonal to these setups since it concurrently tackles multiple heterogeneous and related tasks within each episode.

**Different roles of global and local latent variables.** In this paper, we introduce global latent representations and local latent parameters within a hierarchical architecture. Each type of them plays a distinct role in the proposed model: (1) Global latent representations provide rich task-specific information during the inference of all local latent parameters. This enables the model to generate task-specific decoders to handle heterogeneous tasks in a single episode effectively. (2) Local latent parameters with prediction-aware information constitute task-specific decoders. Each local latent parameter reveals the knowledge corresponding to a specific prediction across different tasks. This enhances the expressive power of the proposed model. In practice, we observe that global latent representations and local latent parameters complement each other when performing predictions in meta-test episodes.

**Advantages of the proposed hierarchical Bayes framework.** We summarize the advantages of the proposed framework. (1) A hierarchical Bayesian framework with global and local latent variables yields a richer and more complex latent space to mitigate the expressiveness bottleneck, thus better parameterizing task-specific functions in stochastic processes. (2) Global and local latent variables capture epistemic uncertainty in representation and parameter levels, respectively, and show improved performance in our experiments.

**Roles of probabilistic HNPs and KL values in meta training.** The probabilistic HNPs encode the context as the heterogeneous prior and reveal the uncertainty resulting from data insufficiency and the extent of observations in tasks. Additionally, minimizing KL terms encourages priors inferred from context sets to stay close to posteriors inferred from target sets, guiding more efficient conditional generation. In training processes, we observed the KL divergence value does not decrease to 0 after convergence, e.g., KL values are in the order of e-1 on Office-home. This is also part of traits in the NPs family, suggesting the approximate posterior and the approximate prior encode different conditional information during the generation of latent variables.

**Real-world examples or benchmarks for episodic multi-task learning.** Episodic multi-task learning has several potential applications in the real world, such as autonomous driving and robotic manipulations. In detail, the autonomous driving system needs to deal with different and related sensor data in an environment. However, the driving environment constantly changes along with the

weather, time, country, etc. Thus, fast adapting of the current multi-tasker to new environments is challenging in this application and our method can be a plausible solution for this challenge.

## B    Properties of Valid Exchangeable Stochastic Processes

Here, we further demonstrate that HNPs are valid stochastic processes, as meeting the *exchangeability* and *marginalization* consistency conditions [18]. As stated in [18]: the conditions, including (finite) exchangeability and marginalization, are sufficient to define a stochastic process with the help of the Kolmogorov extension theorem. Here we follow the notations in the main paper. For convenience, we provide the used symbols and the corresponding descriptions in Table 2.

In this paper, we model the functional posterior distribution of the stochastic process by approximating the joint distribution over all target sets $p(\mathbf{y}_\tau^{1:M}|\mathbf{x}_\tau^{1:M}, \mathcal{C}_\tau^{1:M})$, which is conditioned on all context samples $C_\tau^{1:M}$. To distinguish the order set among different tasks, we use $n_\tau^m$ to denote the number of target samples corresponding to a specific task in the episode $\tau$. For simplicity, we omit the meta-knowledge $\omega$ and $\nu^{1:M}$ in the formulations during the proof.

Table 2: Notations and their corresponding descriptions in this paper.

| Notation | Description |
|---|---|
| $(\cdot)_\tau$ | Variables correspond to an episode. |
| $(\cdot)^m$ | Variables correspond to a single task. |
| $\mathcal{I}_\tau^m$ | A single task in the episode $\tau$, which is sampled from a specific task distribution. |
| $p(\mathcal{I}^m)$ | The specific task distribution. |
| $M$ | The number of task distributions and the number of tasks in a single episode. |
| $\mathcal{I}_\tau^{1:M}$ | All heterogeneous tasks in the episode $\tau$. |
| $\mathcal{C}$ | A context set. |
| $\mathcal{T}$ | A target set. |
| $\bar{x}$ | A context sample feature of the context set. |
| $x$ | A target sample feature of the target set. |
| $\mathbf{x}$ | Set of all target sample features in the target set. |
| $\bar{y}$ | The ground truth of the context sample. |
| $y$ | The ground truth of the target sample. |
| $\mathbf{y}$ | Set of the ground truth of all target samples in the target set. |
| $\mathcal{N}_\mathcal{C}$ | The numbers of context samples in the context set. |
| $\mathcal{N}_\mathcal{T}$ | The numbers of target samples in the target set. |
| $\mathbf{z}_\tau^m$ | The introduced global latent representation for a given task. |
| $\mathbf{w}_{\tau,1:O}^m$ | The introduced local latent parameters for a given task. |

### B.1    Proof of Exchangeability Consistency

We now provide the proof of *Exchangeability Consistency*: the joint prediction distribution is invariant to the permutation of the given multiple tasks and the corresponding samples in each task.

**Theorem B.1.** *(Exchangability) For finite* $n_\tau^* = \sum_{m=1}^M n_\tau^m$, *if* $\pi_\tau^* = \{\pi_\tau^m\}_{m=1}^M$ *is a permutation of* $\{1, \cdots, n^*\}$ *where* $\pi_\tau^m$ *is a permutation of the corresponding order set* $\{1, \cdots, n_\tau^m\}$, *then:*

$$p(\pi_\tau^*(\mathbf{y}_\tau^{1:M})|\pi_\tau^*(\mathbf{x}_\tau^{1:M}), \mathcal{C}_\tau^{1:M}) = p(\mathbf{y}_\tau^{1:M}|\mathbf{x}_\tau^{1:M}, \mathcal{C}_\tau^{1:M}),$$

*where* $\pi_\tau^*(\mathbf{y}_\tau^{1:M}) := (\pi_\tau^1(\mathbf{y}_\tau^1), \cdots, \pi_\tau^M(\mathbf{y}_\tau^M)) = (y_{\tau,\pi_\tau^*(1)}, \cdots, y_{\tau,\pi_\tau^*(n_\tau^*)})$ *and* $\pi_\tau^*(\{\mathbf{x}_\tau^{1:M}\}) := (\pi_\tau^1(\mathbf{x}_\tau^1), \cdots, \pi_\tau^M(\mathbf{x}_\tau^M)) = (x_{\tau,\pi_\tau^*(1)}, \cdots, x_{\tau,\pi_\tau^*(n_\tau^*)})$.

*Proof.*

$$p(\pi_\tau^*(\mathbf{y}_\tau^{1:M})|\pi_\tau^*(\mathbf{x}_\tau^{1:M}), \mathcal{C}_\tau^{1:M})$$

$$= \int\int p(\pi_\tau^*(\mathbf{y}_\tau^{1:M})|\pi_\tau^*(\mathbf{x}_\tau^{1:M}), \mathbf{w}_{\tau,1:O}^{1:M})\Big(\prod_{m=1}^{M} p(\mathbf{w}_{\tau,1:O}^m|\mathbf{z}_\tau^m, \mathcal{C}_\tau^{1:M})\Big)\Big(\prod_{m=1}^{M} p(\mathbf{z}_\tau^m|\mathcal{C}_\tau^m)\Big)d\mathbf{w}_{\tau,1:O}^{1:M}d\mathbf{z}_\tau^{1:M}$$

$$= \int\int \Big(\prod_{i=1}^{n_\tau^*} p(y_{\tau,\pi_\tau^*(i)}|x_{\tau,\pi_\tau^*(i)}, \mathbf{w}_{\tau,1:O}^{1:M})\Big)\Big(\prod_{m=1}^{M} p(\mathbf{w}_{\tau,1:O}^m|\mathbf{z}_\tau^m, \mathcal{C}_\tau^{1:M})\Big)\Big(\prod_{m=1}^{M} p(\mathbf{z}_\tau^m|\mathcal{C}_\tau^m)\Big)d\mathbf{w}_{\tau,1:O}^{1:M}d\mathbf{z}_\tau^{1:M}$$

$$= \int\int \prod_{m=1}^{M}\Big\{\prod_{j=1}^{n_\tau^m} p(y_{\tau,\pi^m(j)}^m|x_{\tau,\pi^m(j)}^m, \mathbf{w}_{\tau,1:O}^m)p(\mathbf{w}_{\tau,1:O}^m|\mathbf{z}_\tau^m, \mathcal{C}_\tau^{1:M})p(\mathbf{z}_\tau^m|\mathcal{C}_\tau^m)\Big\}d\mathbf{w}_{\tau,1:O}^{1:M}d\mathbf{z}_\tau^{1:M}$$

$$= \int\int \prod_{m=1}^{M}\Big\{\prod_{j=1}^{n_\tau^m} p(y_{\tau,j}^m|x_{\tau,j}^m, \mathbf{w}_{\tau,1:O}^m)p(\mathbf{w}_{\tau,1:O}^m|\mathbf{z}_\tau^m, \mathcal{C}_\tau^{1:M})p(\mathbf{z}_\tau^m|\mathcal{C}_\tau^m)\Big\}d\mathbf{w}_{\tau,1:O}^{1:M}d\mathbf{z}_\tau^{1:M}$$

$$= p(\mathbf{y}_\tau^{1:M}|\mathbf{x}_\tau^{1:M}, \mathcal{C}_\tau^{1:M}).$$

□

## B.2 Proof of Marginalization Consistency

We now aim to prove the *Marginalization Consistency* of the proposed model: if marginalizing out a part of the target set in each task, the marginal distribution remains the same as defined on the original target sets without the marginalized part.

**Theorem B.2.** *(Marginalization) Given $\hat{n}_\tau^* = \sum_{m=1}^{M} \hat{n}_\tau^m$, where $1 \leq \hat{n}_\tau^* \leq n_\tau^*$ and for each task $1 \leq \hat{n}_\tau^m \leq n_\tau^m$, the consistency is:*

$$\int p(\mathbf{y}_\tau^{1:M}|\mathbf{x}_\tau^{1:M}, \mathcal{C}_\tau^{1:M})d(\mathbf{y}_\tau^{1:M})_{\hat{n}_\tau^*+1:n_\tau^*} = p((\mathbf{y}_\tau^{1:M})_{1:\hat{n}_\tau^*}|(\mathbf{x}_\tau^{1:M})_{1:\hat{n}_\tau^*}, \mathcal{C}_\tau^{1:M}),$$

*where* $(\mathbf{y}_\tau^{1:M})_{1:\hat{n}_\tau^*} = ((\mathbf{y}_\tau^1)_{1:\hat{n}_\tau^1}, \cdots, (\mathbf{y}_\tau^M)_{1:\hat{n}_\tau^M}) = (y_{\tau,1}, \cdots, y_{\tau,\hat{n}_\tau^*})$ *and* $(\mathbf{x}_\tau^{1:M})_{1:\hat{n}_\tau^*} = ((\mathbf{x}_\tau^1)_{1:\hat{n}_\tau^1}, \cdots, (\mathbf{x}_\tau^M)_{1:\hat{n}_\tau^M}) = (x_{\tau,1}, \cdots, x_{\tau,\hat{n}_\tau^*}).$

*Proof.*

$$\int p(\mathbf{y}_\tau^{1:M}|\mathbf{x}_\tau^{1:M}, \mathcal{C}_\tau^{1:M})d(\mathbf{y}_\tau^{1:M})_{\hat{n}_\tau^*+1:n_\tau^*}$$

$$= \int\int\int p(\mathbf{y}_\tau^{1:M}|\mathbf{x}_\tau^{1:M}, \mathbf{w}_{\tau,1:O}^{1:M})\Big(\prod_{m=1}^{M} p(\mathbf{w}_{\tau,1:O}^m|\mathbf{z}_\tau^m, \mathcal{C}_\tau^{1:M})\Big)\Big(\prod_{m=1}^{M} p(\mathbf{z}_\tau^m|\mathcal{C}_\tau^m)\Big)d\mathbf{w}_{\tau,1:O}^{1:M}d\mathbf{z}_\tau^{1:M}d(\mathbf{y}_\tau^{1:M})_{\hat{n}_\tau^*+1:n_\tau^*}$$

$$= \int\int\int \Big(\prod_{i=1}^{n_\tau^*} p(y_{\tau,i}|x_{\tau,i}, \mathbf{w}_{\tau,1:O}^{1:M})\Big)\Big(\prod_{m=1}^{M} p(\mathbf{w}_{\tau,1:O}^m|\mathbf{z}_\tau^m, \mathcal{C}_\tau^{1:M})\Big)\Big(\prod_{m=1}^{M} p(\mathbf{z}_\tau^m|\mathcal{C}_\tau^m)\Big)d\mathbf{w}_{\tau,1:O}^{1:M}d\mathbf{z}_\tau^{1:M}d(\mathbf{y}_\tau^{1:M})_{\hat{n}_\tau^*+1:n_\tau^*}$$

$$= \int\int \prod_{m=1}^{M}\Big\{\int\prod_{j=1}^{n_\tau^m} p(y_{\tau,j}^m|x_{\tau,j}^m, \mathbf{w}_{\tau,1:O}^m)p(\mathbf{w}_{\tau,1:O}^m|\mathbf{z}_\tau^m, \mathcal{C}_\tau^{1:M})p(\mathbf{z}_\tau^m|\mathcal{C}_\tau^m)d(\mathbf{y}_\tau^m)_{\hat{n}_\tau^m+1:n_\tau^m}\Big\}d\mathbf{w}_{\tau,1:O}^{1:M}d\mathbf{z}_\tau^{1:M}$$

$$= \int\int \prod_{m=1}^{M}\Big\{\prod_{j=1}^{\hat{n}_\tau^m} p(y_{\tau,j}^m|x_{\tau,j}^m, \mathbf{w}_{\tau,1:O}^m)p(\mathbf{w}_{\tau,1:O}^m|\mathbf{z}_\tau^m, \mathcal{C}_\tau^{1:M})p(\mathbf{z}_\tau^m|\mathcal{C}_\tau^m)$$

$$\int\prod_{j=\hat{n}_\tau^m+1}^{n_\tau^m} p(y_{\tau,j}^m|x_{\tau,j}^m, \mathbf{w}_{\tau,1:O}^m)d(\mathbf{y}_\tau^m)_{\hat{n}_\tau^m+1:n_\tau^m}\Big\}d\mathbf{w}_{\tau,1:O}^{1:M}d\mathbf{z}_\tau^{1:M}$$

$$= \int\int \prod_{m=1}^{M}\Big\{\prod_{j=1}^{\hat{n}_\tau^m} p(y_{\tau,j}^m|x_{\tau,j}^m, \mathbf{w}_{\tau,1:O}^m)p(\mathbf{w}_{\tau,1:O}^m|\mathbf{z}_\tau^m, \mathcal{C}_\tau^{1:M})p(\mathbf{z}_\tau^m|\mathcal{C}_\tau^m)\Big\}d\mathbf{w}_{\tau,1:O}^{1:M}d\mathbf{z}_\tau^{1:M}$$

$$= p((\mathbf{y}_\tau^{1:M})_{1:\hat{n}_\tau^*}|(\mathbf{x}_\tau^{1:M})_{1:\hat{n}_\tau^*}, \mathcal{C}_\tau^{1:M}).$$

□

## C  Tractable and Scalable Optimization

For the proposed HNPs, it is intractable to obtain the true joint posterior $p(\mathbf{w}_{\tau,1:O}^{1:M}, \mathbf{z}_\tau^{1:M} | \mathcal{T}_\tau^{1:M}; \omega, \nu^{1:M})$ for each episode. Thus, we employ variational inference to optimize the designed model by approximating the true joint posterior in each episode. To do so, we introduce the variational joint posterior distribution:

$$q_{\theta,\phi}(\mathbf{w}_{\tau,1:O}^{1:M}, \mathbf{z}_\tau^{1:M} | \mathcal{T}_\tau^{1:M}; \omega, \nu^{1:M}) = \prod_{m=1}^{M} q_\theta(\mathbf{z}_\tau^m | \mathcal{T}_\tau^m; \nu^m) q_\phi(\mathbf{w}_{\tau,1:O}^m | \mathbf{z}_\tau^m, \mathcal{T}_\tau^{1:M}; \omega), \quad (1)$$

where $q_\theta(\mathbf{z}_\tau^m | \mathcal{T}_\tau^m; \nu^m)$ and $q_\phi(\mathbf{w}_{\tau,1:O}^m | \mathbf{z}_\tau^m, \mathcal{T}_\tau^{1:M}; \omega)$ are variational posteriors of their corresponding latent variables. We note that variational posteriors are inferred from the target sets that are available in the meta-training stage. The variational posteriors are parameterized as diagonal Gaussian distributions [44]. The inference networks $\theta$ and $\phi$ are shared by the variational posteriors and their corresponding priors, following the protocol of vanilla NPs [18].

### C.1  Derivation of Approximate ELBO for HNPs

By incorporating the variational posteriors in Eq. (1) into the modeling of HNPs in the main paper, we can derive the approximate ELBO $L_{\mathrm{HNPs}}(\omega, \nu^{1:M}, \theta, \phi)$ as follows:

$$\log p(\mathcal{T}_\tau^{1:M} | \mathcal{C}_\tau^{1:M}; \omega, \nu^{1:M})$$

$$= \sum_{m=1}^{M} \log p(\mathcal{T}_\tau^m | \mathcal{C}_\tau^{1:M}; \omega, \nu^m)$$

$$= \sum_{m=1}^{M} \left\{ \log \int \left\{ \int p(\mathbf{y}_\tau^m | \mathbf{x}_\tau^m, \mathbf{w}_{\tau,1:O}^m) p_\phi(\mathbf{w}_{\tau,1:O}^m | \mathbf{z}_\tau^m, \mathcal{C}_\tau^{1:M}; \omega) d\mathbf{w}_{\tau,1:O}^m \right\} p_\theta(\mathbf{z}_\tau^m | \mathcal{C}_\tau^m; \nu^m) d\mathbf{z}_\tau^m \right\}$$

$$= \sum_{m=1}^{M} \left\{ \log \int \left\{ \int p(\mathbf{y}_\tau^m | \mathbf{x}_\tau^m, \mathbf{w}_{\tau,1:O}^m) p_\phi(\mathbf{w}_{\tau,1:O}^m | \mathbf{z}_\tau^m, \mathcal{C}_\tau^{1:M}; \omega) \frac{q_\phi(\mathbf{w}_{\tau,1:O}^m | \mathbf{z}_\tau^m, \mathcal{T}_\tau^{1:M}; \omega)}{q_\phi(\mathbf{w}_{\tau,1:O}^m | \mathbf{z}_\tau^m, \mathcal{T}_\tau^{1:M}; \omega)} d\mathbf{w}_{\tau,1:O}^m \right\} \right.$$

$$\left. p_\theta(\mathbf{z}_\tau^m | \mathcal{C}_\tau^m; \nu^m) \frac{q_\theta(\mathbf{z}_\tau^m | \mathcal{T}_\tau^m; \nu^m)}{q_\theta(\mathbf{z}_\tau^m | \mathcal{T}_\tau^m; \nu^m)} d\mathbf{z}_\tau^m \right\}$$

$$\geq \sum_{m=1}^{M} \left\{ \mathbb{E}_{q_\theta(\mathbf{z}_\tau^m | \mathcal{T}_\tau^m; \nu^m)} \left\{ \log \int p(\mathbf{y}_\tau^m | \mathbf{x}_\tau^m, \mathbf{w}_{\tau,1:O}^m) p_\phi(\mathbf{w}_{\tau,1:O}^m | \mathbf{z}_\tau^m, \mathcal{C}_\tau^{1:M}; \omega) \frac{q_\phi(\mathbf{w}_{\tau,1:O}^m | \mathbf{z}_\tau^m, \mathcal{T}_\tau^{1:M}; \omega)}{q_\phi(\mathbf{w}_{\tau,1:O}^m | \mathbf{z}_\tau^m, \mathcal{T}_\tau^{1:M}; \omega)} d\mathbf{w}_{\tau,1:O}^m \right\} \right.$$

$$\left. - \mathbb{D}_{\mathrm{KL}}[q_\theta(\mathbf{z}_\tau^m | \mathcal{T}_\tau^m; \nu^m) || p_\theta(\mathbf{z}_\tau^m | \mathcal{C}_\tau^m; \nu^m)] \right\}$$

$$\geq \sum_{m=1}^{M} \left\{ \mathbb{E}_{q_\theta(\mathbf{z}_\tau^m | \mathcal{T}_\tau^m; \nu^m)} \left\{ \mathbb{E}_{q_\phi(\mathbf{w}_{\tau,1:O}^m | \mathbf{z}_\tau^m, \mathcal{T}_\tau^{1:M}; \omega)} [\log p(\mathbf{y}_\tau^m | \mathbf{x}_\tau^m, \mathbf{w}_{\tau,1:O}^m)] \right. \right.$$

$$\left. - \mathbb{D}_{\mathrm{KL}}[q_\phi(\mathbf{w}_{\tau,1:O}^m | \mathbf{z}_\tau^m, \mathcal{T}_\tau^{1:M}; \omega) || p_\phi(\mathbf{w}_{\tau,1:O}^m | \mathbf{z}_\tau^m, \mathcal{C}_\tau^{1:M}; \omega)] \right\}$$

$$\left. - \mathbb{D}_{\mathrm{KL}}[q_\theta(\mathbf{z}_\tau^m | \mathcal{T}_\tau^m; \nu^m) || p_\theta(\mathbf{z}_\tau^m | \mathcal{C}_\tau^m; \nu^m)] \right\} = L_{\mathrm{HNPs}}(\omega, \nu^{1:M}, \theta, \phi).$$

$$(2)$$

In general, when the variational joint posterior is flexible enough, the posterior approximation gap between the variational posterior and the intractable true posterior can be reduced to an arbitrarily small quantity [45]. In this case, maximizing the approximate ELBO increases the overall log likelihood in the proposed model accordingly. We construct task-specific decoders in a data-driven way by inferring local latent parameters $\mathbf{w}_{1:O}^m$ from all context sets and the meta-knowledge. This enables our model to amortize the training cost of each task-specific decoder, further reducing the model's over-fitting behaviors for episodic multi-task learning.

## C.2 Meta-Training Objective

In practice, we consider the loss function as the negative approximate ELBO of HNPs as given in Eq. (2). By adopting Monte Carlo sampling [44, 46], the meta-training objective for each episode is:

$$
\begin{aligned}
- L_{\text{HNPs}}(\omega, \nu^{1:M}, \theta, \phi) \approx \sum_{m=1}^{M} \Bigg\{ & \frac{1}{N_z} \sum_{i=1}^{N_z} \Big\{ \frac{1}{N_w} \sum_{j=1}^{N_w} [- \log p(\mathbf{y}_\tau^m | \mathbf{x}_\tau^m, \mathbf{w}_{\tau,1:O}^{m\,(j)})] \\
& + \mathbb{D}_{\text{KL}}[q_\phi(\mathbf{w}_{\tau,1:O}^m | \mathbf{z}_\tau^{m\,(i)}, \mathcal{T}_\tau^{1:M}; \omega) || p_\phi(\mathbf{w}_{\tau,1:O}^m | \mathbf{z}_\tau^{m\,(i)}, \mathcal{C}_\tau^{1:M}; \omega)] \Big\} \\
& + \mathbb{D}_{\text{KL}}[q_\theta(\mathbf{z}_\tau^m | \mathcal{T}_\tau^m; \nu^m) || p_\theta(\mathbf{z}_\tau^m | \mathcal{C}_\tau^m; \nu^m)] \Bigg\},
\end{aligned}
\tag{3}
$$

where $\mathbf{z}_\tau^{m\,(i)}$ and $\mathbf{w}_{\tau,1:O}^{m\,(j)}$ are sampled from their corresponding variational posteriors. $N_z$ and $N_w$ are the number of Monte Carlo samples for $\mathbf{z}_\tau^m$ and $\mathbf{w}_{\tau,1:O}^m$, respectively.

## C.3 Meta-Test Prediction

At the meta-test stage, we perform predictions with the learned model on the target sets for a new episode $\tau^*$, which involves the prior distributions of global and local latent variables. We again approximate the predictive distribution with Monte Carlo estimates:

$$
p(\mathcal{T}_{\tau^*}^{1:M} | \mathcal{C}_{\tau^*}^{1:M}; \omega, \nu^{1:M}) \approx \prod_{m=1}^{M} \Bigg\{ \frac{1}{N_z} \sum_{i=1}^{N_z} \frac{1}{N_w} \sum_{j=1}^{N_w} p(\mathbf{y}_{\tau^*}^m | \mathbf{x}_{\tau^*}^m, \mathbf{w}_{\tau^*,1:O}^{m\,(j)}) \Bigg\},
\tag{4}
$$

where $\mathbf{w}_{\tau^*,1:O}^{m\,(j)} \sim p_\phi(\mathbf{w}_{\tau^*,1:O}^m | \mathbf{z}_{\tau^*}^{m\,(i)}, \mathcal{C}_{\tau^*}^{1:M}; \omega)$ and $\mathbf{z}_{\tau^*}^{m\,(i)} \sim p_\theta(\mathbf{z}_{\tau^*}^m | \mathcal{C}_{\tau^*}^m; \nu^m)$. Here the Monte Carlo samples follow their corresponding prior distributions since the target sets are unavailable during the meta-test.

# D More Experimental Details

## D.1 Transformer-structured Inference Modules in Regression Scenarios

Here we present the implementation details of transformer-structured inference module $\theta$ in regression scenarios. In a single episode $\tau$, the module $\theta$ encodes the task-specific information into each refined task-specific token $\nu^m$, and then infers the prior distribution or the variational posterior distribution for the global latent representation $\mathbf{z}_\tau^m$. For episodic multi-task regression, the local latent parameters $\mathbf{w}_{\tau,1:O}^m$ construct a task-specific regressor during inference. We assume that the output of the decoder follows a Gaussian distribution for regression tasks. Thus, $\mathbf{w}_{\tau,1:O}^m$ are instantiated as parameters for generating the mean and variance of predictions.

## D.2 Backbone and Training Details

Following the protocol of [9], we apply the pre-trained deep model as the backbone for the proposed method and baselines to extract the input features under the episodic multi-task classification setup. To be specific, we adopt VGGnet [47] for `Office-Home`, and its feature size is $4096$. We take ResNet18 [48] for `DomainNet` with input size $512$. In practice, we train our method and baselines by the Adam optimizer [49] using an NVIDIA Tesla V100 GPU. The learning rate is initially set as $1e-4$ and decreases with a factor of $0.5$ every $3,000$ iterations.

## D.3 Implementation Details

To clearly show implementation details for episodic multi-task classification, we attach the Python code of the proposed HNPs in the following.

```python
import torch
import torch.nn as nn

from torch.distributions import Normal
from torch.distributions.kl import kl_divergence
```

```python
import basic_model

class HNP(nn.Module):
    def __init__(self, config):
        super(HNP, self).__init__()

        self.dataset = config["dset_name"]
        self.num_task = config["num_task"]
        self.way_number = config["way_number"]
        self.shot_number = config["shot_number"]
        self.w_repeat = config["w_repeat"]
        self.z_repeat = config["z_repeat"]

        if self.dataset == "domainnet":
            self.x_feature = 512
        else:
            self.x_feature = 4096
        self.d_feature = self.x_feature
        config["d_feature"] = self.d_feature

        model = basic_model
        # task-wise and class-wise transformers
        self.task_probabilistic_encoder = model.ProbabilisticEncoder_theta(
            config, self.d_feature, self.num_task)
        self.class_probabilistic_encoder = model.ProbabilisticEncoder_phi(
            config, self.d_feature, self.way_number)

    @staticmethod
    def print_parameters(net):
        for name, parameters in net.named_parameters():
            print(name, ':', parameters.size())

    ####################################################################
    # Transformer inference for global latent representations
    ####################################################################
    def task_wise_transformer_inference(self, x_c_order, x_t_order):

        task_prior = x_c_order.view(self.num_task, -1, self.d_feature)
        task_posterior = x_t_order.view(self.num_task, -1, self.d_feature)

        # infer for prior
        z_pmu, z_psigma = self.task_probabilistic_encoder(task_prior)
        z_pdistirbution = Normal(z_pmu, z_psigma)
        z_psample = z_pdistirbution.rsample([self.a_z])

        # infer for posterior
        z_qmu, z_qsigma = self.task_probabilistic_encoder(task_posterior)
        z_qdistirbution = Normal(z_qmu, z_qsigma)
        z_qsample = z_qdistirbution.rsample([self.z_repeat])

        # kl_z
        kl_z = kl_divergence(z_qdistirbution, z_pdistirbution).sum(dim=1)
        return kl_z, z_psample, z_qsample
```

```python
        ######################################################################
        # Transformer inference for local latent parameters
        ######################################################################
        def class_wise_transformer_inference(self, a_qsample, a_psample,
                                             x_c_order, x_t_order, x_t):

            kl_w = []
            output_prior_list = []
            output_posterior_list = []
            for num in range(self.num_task):

                if self.training:
                    task_embedding = a_qsample[:, num, :]
                else:
                    task_embedding = a_psample[:, num, :]

                class_prior = x_c_order
                class_posterior = x_t_order

                # infer for prior
                phi_pmu, phi_psigma = self.class_probabilistic_encoder(class_prior,
                                                                       task_embedding)
                phi_pdistirbution = Normal(phi_pmu, phi_psigma)

                # infer for posterior
                phi_qmu, phi_qsigma = self.class_probabilistic_encoder(class_posterior,
                                                                       task_embedding)
                phi_qdistirbution = Normal(phi_qmu, phi_qsigma)

                # task-specific kl_w
                task_specific_kl_w = kl_divergence(phi_qdistirbution,
                                                   phi_pdistirbution).mean(dim=0).sum()
                kl_w.append(task_specific_kl_w.view(1))

                # perform prediction
                predict_samples = x_t[num].contiguous().view(-1, self.d_feature)
                repeat_predict_samples = predict_samples.unsqueeze(0)
                repeat_predict_samples = repeat_predict_samples.expand(
                    self.w_repeat * self.z_repeat, predict_samples.shape[0],
                    predict_samples.shape[1]).contiguous()

                if self.training:
                    phi_qsample = phi_qdistirbution.rsample([self.w_repeat])
                    phi_qsample = phi_qsample.transpose(0, 1)
                    phi_qsample = phi_qsample.reshape(self.way_number,-1,
                    self.d_feature)
                    classifier_q = phi_qsample.transpose(0, 1).transpose(1, 2)

                    phi_psample = phi_pdistirbution.rsample([self.w_repeat])
                    phi_psample = phi_psample.transpose(0, 1)
                    phi_psample = phi_psample.reshape(self.way_number,-1,
                    self.d_feature)
```

```python
113                        classifier_p = phi_psample.transpose(0, 1).transpose(1, 2)
114
115                else:
116                        classifier_q = phi_qmu.unsqueeze(1).repeat(1, self.w_repeat, 1, 1)
117                        classifier_q = classifier_q.reshape(self.way_number, -1,
118                                                             self.d_feature)
119                        classifier_q = classifier_q.transpose(0, 1).transpose(1, 2)
120
121                        classifier_p = phi_pmu.unsqueeze(1).repeat(1, self.w_repeat, 1, 1)
122                        classifier_p =classifier_p.reshape(self.way_number, -1,
123                                                           self.d_feature)
124                        classifier_p = classifier_p.transpose(0, 1).transpose(1, 2)
125
126                output_posterior = torch.bmm(repeat_predict_samples, classifier_q)
127                output_posterior = output_posterior.unsqueeze(0)
128                output_prior = torch.bmm(repeat_predict_samples, classifier_p)
129                output_prior = output_prior.unsqueeze(0)
130                output_posterior_list.append(output_posterior)
131                output_prior_list.append(output_prior)
132
133            kl_w = torch.cat(kl_w, 0)
134            output_posterior_all = torch.cat(output_posterior_list, 0)
135            output_prior_all = torch.cat(output_prior_list, 0)
136            return kl_w, output_posterior_all, output_prior_all
137
138        def forward(self, inputs_batch, labels_batch):
139            # dividing the context and target samples
140            label_all = labels_batch.squeeze(-1)
141            _, indices = torch.sort(label_all[0, :, 0])
142            x_c = inputs_batch[:, :, :self.shot_number, :]
143            x_t = inputs_batch[:, :, self.shot_number:, :]
144            x_c_order = x_c[:, indices, :, :]
145            x_t_order = x_t[:, indices, :, :]
146
147            # task-wise transformer
148            kl_z, z_psample, z_qsample = self.task_wise_transformer_inference(
149                x_c_order, x_t_order)
150            kl_w, output_posterior_all, output_prior_all \
151                = self.class_wise_transformer_inference(z_qsample, z_psample,
152                                                        x_c_order, x_t_order, x_t)
153            if self.training:
154                return output_posterior_all, output_prior_all, kl_z, kl_w
155            else:
156                return output_prior_all
```

We also provide implementation details about the introduced transformer-structured inference modules for hierarchical latent variables as follows.

```python
1   import torch
2   import torch.nn as nn
3   import torch.nn.functional as f
4   from torch.nn import TransformerEncoder, TransformerEncoderLayer
```

```python
class ProbabilisticEncoder_theta(nn.Module):
    def __init__(self, config, feature_dim, num_learnable_token):
        super(ProbabilisticEncoder_theta, self).__init__()
        # inducing tokens
        self.learnable_tokens = \
            nn.Parameter(torch.empty((num_learnable_token, 1, feature_dim),
                         dtype=torch.float32).normal_(0.,0.1),requires_grad=True)

        emsize = feature_dim
        ninp = emsize
        nhead = int(emsize/64)
        nhid = emsize
        task_encoder_layer = TransformerEncoderLayer(ninp, nhead, nhid, 0.2,
                             activation='gelu', batch_first=True)
        self.transformer_encoder = TransformerEncoder(task_encoder_layer, 1)
        sizes = [emsize, emsize, emsize, emsize]
        self.mu_infer = nn.Linear(sizes[-2], sizes[-1])
        self.sigma_infer = nn.Linear(sizes[-2], sizes[-1])

    def forward(self, input):
        context = input
        query = self.learnable_tokens

        # transformer, interactions between tokens
        src = torch.cat([context, query], 1)
        updated_src = self.transformer_encoder(src)
        n_all, n_context = src.shape[1], context.shape[1]
        updated_learnable_tokens = updated_src[:, n_context:, :]

        # amortization inference
        x = updated_learnable_tokens.squeeze(1)
        mu = self.mu_infer(x)
        sigma = f.softplus(self.sigma_infer(x), beta=1, threshold=20)
        return mu, sigma

class ProbabilisticEncoder_phi(nn.Module):
    def __init__(self, config, feature_dim, num_learnable_token):
        super(ProbabilisticEncoder_phi, self).__init__()
        self.way_number = config["way_number"]
        self.d_feature = config["d_feature"]
        # inducing tokens
        self.learnable_tokens = \
            nn.Parameter(torch.empty((num_learnable_token, 1, feature_dim),
                         dtype=torch.float32).normal_(0., 0.1),requires_grad=True)

        emsize = feature_dim
        ninp = emsize
        nhead = int(emsize / 64)
        nhid = emsize
        task_encoder_layer = TransformerEncoderLayer(ninp, nhead, nhid, 0.2,
                             activation='gelu', batch_first=True)
        self.transformer_encoder = TransformerEncoder(task_encoder_layer, 1)
```

```
59          sizes = [emsize, emsize, emsize, emsize]
60          self.mu_infer = nn.Linear(sizes[-2], sizes[-1])
61          self.sigma_infer = nn.Linear(sizes[-2], sizes[-1])
62
63      def forward(self, input, embedding):
64          context = input.transpose(0, 1).reshape(self.way_number, -1, self.d_feature)
65          query = self.learnable_tokens
66          src = torch.cat([context, query], 1)
67          updated_src = self.transformer_encoder(src)
68          n_all, n_context = src.shape[1], context.shape[1]
69          updated_learnable_tokens = updated_src[:, n_context:, :]
70
71          # amortization inference
72          x = updated_learnable_tokens + embedding
73          mu = self.mu_infer(x)
74          sigma = f.softplus(self.sigma_infer(x), beta=1, threshold=20)
75          return mu, sigma
```

# E    Algorithm of the proposed HNPs

---

**Algorithm 1:** Meta-training phase of HNPs.

---

**Input**  : $M$ distinct and relevant task distributions $p(\mathcal{I}^{1:M})$, numbers of Monte Carlo samples $N_z$ and $N_w$, learning rates $\alpha$.

**Output**: Meta-trained transformer inference module $\theta$ and $\phi$, learnable tokens $\omega_{1:O}$ and $\nu^{1:M}$.

Initialize transformer inference module $\{\theta, \nu^{1:M}\}$ ;

Initialize transformer inference module $\{\phi, \omega_{1:O}\}$ ;

**while** *Meta-Training not Completed* **do**

    // sample a meta-training episode indexed with $\tau$.

    **for** $m = 1$ *to* $M$ **do**

        Sample a task $\mathcal{I}_\tau^m \sim p(\mathcal{I}^m)$, which shares the same target space $\mathcal{Y}_\tau$ with other tasks in the episode;

        Sample a context set $\mathcal{C}_\tau^m$ and a target set $\mathcal{T}_\tau^m$ for the task $\mathcal{I}_\tau^m$;

    **end**

    // infer hierarchical latent variables.

    **for** $m = 1$ *to* $M$ **do**

        Apply transformer inference module $\theta$ to infer prior and variational posterior distributions over $\mathbf{z}_\tau^m$ as Eq. (5), Eq. (6), and Eq. (7) in the main paper;

        Draw $N_z$ samples from the variational posterior, $\{\mathbf{z}_\tau^{m\,(i)}\}_{i=1}^{N_z}$;

        **for** $i = 1$ *to* $N_z$ **do**

            **for** $o = 1$ *to* $O$ **do**

                Apply transformer inference module $\phi$ to infer prior and variational posterior distributions over $\mathbf{w}_{\tau,o}^m$ as Eq. (8), Eq. (9), and Eq. (10) in the main paper;

                Draw $N_w$ samples from the variational posterior, $\{\mathbf{w}_{\tau,o}^{m\,(j)}\}_{j=1}^{N_w}$;

            **end**

        **end**

    **end**

    // optimize the objective.

    Compute predictive distributions and minimize the empirical objective in Eq. (3) ;

    Update $\theta$, $\phi$, $\omega_{1:O}$ and $\nu^{1:M}$ with learning rate $\alpha$.

**end**

---

**Algorithm 2:** Meta-test phase of HNPs.

---

**Input** : Meta-trained transformer inference module $\theta$ and $\phi$, learned tokens $\omega_{1:O}$ and $\nu^{1:M}$, numbers of Monte Carlo samples $N_z$ and $N_w$.
**Output** : Prediction results.
**while** *Meta-Test not Completed* **do**
    `// given a meta-test episode indexed with` $\tau^*$`.`
    Collect the context sets of all tasks in the episode, $\mathcal{C}_{\tau^*}^{1:M}$.
    **for** $m = 1$ *to* $M$ **do**
        Apply transformer inference module $\theta$ to infer prior distributions over $\mathbf{z}_{\tau^*}^m$ as Eq. (5), Eq. (6), and Eq. (7) in the main paper;
        Draw $N_z$ samples from the prior distribution, $\{\mathbf{z}_{\tau^*}^{m\,(i)}\}_{i=1}^{N_z}$;
        **for** $i = 1$ *to* $N_z$ **do**
            **for** $o = 1$ *to* $O$ **do**
                Apply transformer inference module $\phi$ to infer the prior distribution over $\mathbf{w}_{\tau^*,o}^m$ as Eq. (8), Eq. (9), and Eq. (10) in the main paper;
                Draw $N_w$ samples from the prior distribution, $\{\mathbf{w}_{\tau^*,o}^{m\,(j)}\}_{j=1}^{N_w}$;
            **end**
        **end**
    **end**
    Perform predictions on the target sets in Eq. (4) ;
**end**

---

Table 3: **Performance comparisons on Office-Home under the Xtask5way1shot setup during inference.**

| Number of tasks | 1 | 2 | 3 | 4 |
|---|---|---|---|---|
| Average accuracy | $64.33 \pm 0.85$ | $70.75 \pm 0.67$ | $74.95 \pm 0.57$ | $76.29 \pm 0.51$ |

# F   More Experimental Results under Episodic Multi-Task Setup

## F.1   Effects of More "Tasks"

To investigate the effects of more "tasks" in the episodic multi-task setup, we perform experiments on Office-Home under the Xway5way1shot setup by gradually increasing tasks during inference. Table 3 shows that the average accuracy increases with more tasks. The main reason is that more tasks can provide richer transferable information. Our model benefits from the positive transfer among tasks, and thus obtaining higher performance gain from more tasks.

## F.2   4task20way1shot v.s. 20way4shot

To show the effectiveness of the proposed method, we conduct experiments with the 20-way 4-shot setup, which needs to mix samples from all tasks in one episode. As shown in Table 4, MAML and Proto.Net perform better under the 20-way 4-shot but cannot outperform our method. The main reason is that our method can better handle distribution shifts among tasks by exploring task-relatedness rather than simply mixing them together.

## F.3   Comparisons with More Recent Works

To compare the proposed method with more recent works, we perform experiments on Office-Home under the 4task5way1shot and 4task5way5shot setups. We provide some brief descriptions of two recent works as follows:

[50] theoretically addresses the conclusion that MTL methods are powerful and efficient alternatives to GBML for meta-learning applications. However, our method inherits the advantages of multi-task learning and meta-learning: simultaneously generalizing meta-knowledge from past to new episodes and exploiting task-relatedness across heterogeneous tasks in every single episode. Thus, our method is more suitable for solving the data-insufficiency problem.

Table 4: **Performance comparisons under the 4task20way1shot and 20way4shot setups.**

| Methods | 4task20way1shot | 20way4shot |
|---------|-----------------|------------|
| MAML | $34.29 \pm 0.19$ | $37.23 \pm 0.25$ |
| Proto.Net | $32.72 \pm 0.18$ | $37.12 \pm 0.22$ |
| Ours | $51.82 \pm 0.23$ | - |

Table 5: **Performance comparisons with more recent baselines.**

| Methods | 1-shot | 5-shot |
|---------|--------|--------|
| MTL-bridge [50] | $64.31 \pm 0.55$ | $75.10 \pm 0.51$ |
| MLTI [51] | $70.69 \pm 0.73$ | $79.59 \pm 0.58$ |
| Ours | $76.29 \pm 0.51$ | $80.80 \pm 0.42$ |

[51] augments the task set in meta-learning through interpolation. Our method fully utilizes several observed tasks in a single episode rather than generating additional tasks.

Table 5 shows that our method significantly outperforms other baselines with severely insufficient data, such as 1-shot. This is consistent with the conclusion obtained in the main paper.

### F.4 Comparisons on Another Benchmark Dataset

We validate the performance of methods on the Office31 dataset [52, 53] under the 3task5way1shot setup. This dataset contains 31 object categories in three domains: Amazon, DSLR, and Webcam. Table 6 shows our method outperforms baseline methods, demonstrating our model's effectiveness in addressing the data insufficiency under the episodic setup.

## G  More Experimental Results under Conventional Multi-Task Setup

To show comparisons with existing multi-task models, which are designed for conventional multi-task learning, we extend the proposed HNPs to conventional multi-task learning settings for both regression and classification tasks by considering only one episode during training and inference. Under conventional multi-task settings (MIMO), we investigate their effectiveness in exploring shared knowledge when limited tasks and samples are available during training.

### G.1  Conventional Multi-Task Regression

**Dataset and Settings.**   We show the effectiveness of HNPs for conventional multi-task regression. We design experiments for rotation angle estimation on the `Rotated MNIST` dataset [54]. Each task is an angle estimation problem for a digit, and different tasks corresponding to individual digits are related because they share the same rotation angle space. Each image is rotated by $0°$ through $90°$ in intervals of $10°$, where the rotation angle is the regression target. We randomly choose $0.1\%$ training samples per task per angle as the training set.

We use the average of normalized mean squared errors (NMSE) of all tasks as the performance measurement. The lower NMSE, the better the performance. We provide $95\%$ confidence intervals for the errors from five runs. Descriptions of baselines can be found in Section G.2.

**Results and Discussions.**   The experimental results are summarized in Table 7. The proposed HNPs outperform other counterpart methods by yielding a lower mean error. This demonstrates the effectiveness of HNPs in capturing task-relatedness for improved regression performance.

### G.2  Conventional Multi-Task Classification

**Datasets and Settings.**   `Office-Caltech` [56] contains ten categories shared between Office-31 [52] and Caltech-256 [57]. One task uses data from Caltech-256, and the other tasks use data from Office-31, whose images were collected from three distinct domains/tasks, namely Amazon, Webcam

Table 6: **Performance comparisons on the Office31 dataset.**

| Methods | ERM | Proto.Net | CNPs | NPs | TNP-D | Ours |
|---|---|---|---|---|---|---|
| Average Accuracy | $63.53 \pm 0.71$ | $64.54 \pm 0.64$ | $49.02 \pm 0.74$ | $40.52 \pm 0.75$ | $69.69 \pm 0.87$ | $71.89 \pm 0.52$ |

Table 7: **Conventional multi-task regression (normalized mean squared errors) for rotation angle estimation.**

| Methods | Average NMSE |
|---|---|
| STL | $.215 \pm .001$ |
| VSTL | $.224 \pm .004$ |
| BMTL | $.118 \pm .003$ |
| VBMTL | $.121 \pm .003$ |
| LearnToBranch [55] | $.109 \pm .002$ |
| VMTL [8] | $.110 \pm .003$ |
| HNPs | $\mathbf{.103} \pm .001$ |

and DSLR. There are $8 \sim 151$ samples per category per task and $2,533$ images. `ImageCLEF` [9], the benchmark for the ImageCLEF domain adaptation challenge, contains 12 common categories shared by four public datasets/tasks: Caltech-256, ImageNet ILSVRC 2012, Pascal VOC 2012, and Bing. There are $2,400$ images in total. `Office-Home` [58] mentioned in the main paper is also used under this setting.

We adopt the standard evaluation protocols [9] for multi-task classification datasets. We randomly select 5%, 10% and 20% labeled data for training, which correspond to about 3, 6 and 12 samples per category per task, respectively. In this case, each task has insufficient training data for building a reliable classifier without overfitting. The average accuracy of all tasks is used for measuring the overall performance. We again provide 95% confidence intervals for the errors from five runs.

**Alternatives Methods.** We conduct a thorough comparison with alternative multi-task learning models. Single-task learning (STL) is implemented by task-specific feature extractors and predictors without knowledge sharing among tasks. Basic multi-task learning (BMTL) shares feature extractors and adds task-specific predictors. We also define variational extensions of single-task learning (VSTL) and basic multi-task learning (VBMTL), which treat predictors as latent variables [8]. For a fair comparison, all the baseline methods mentioned above share the same architecture of the feature extractor and the train-test splits. We also compare the proposed HNPs to representative multi-task models. MTL-Uncertainty [1], MRN [9], LearnToBranch [55] are deep MTL methods, employing deep neural networks to construct information-sharing mechanisms for tasks. TCGBML [59], MTVIB [60] and VMTL [8] are probabilistic MTL methods, applying Bayes frameworks to model the relationships among tasks.

Table 8: **Classification performance (average accuracy) on** `Office-Home`**,** `Office-Caltech` **and** `ImageCLEF`**.**

| Methods | Office-Home | | | Office-Caltech | | | ImageCLEF | | |
|---|---|---|---|---|---|---|---|---|---|
| | 5% | 10% | 20% | 5% | 10% | 20% | 5% | 10% | 20% |
| STL | $49.2 \pm 0.2$ | $58.3 \pm 0.1$ | $64.9 \pm 0.1$ | $88.6 \pm 0.3$ | $90.7 \pm 0.2$ | $92.4 \pm 0.3$ | $62.6 \pm 0.2$ | $69.7 \pm 0.3$ | $76.2 \pm 0.3$ |
| VSTL | $51.1 \pm 0.1$ | $60.2 \pm 0.2$ | $65.8 \pm 0.2$ | $89.0 \pm 0.2$ | $91.1 \pm 0.2$ | $93.4 \pm 0.3$ | $64.9 \pm 0.3$ | $70.8 \pm 0.3$ | $77.2 \pm 0.2$ |
| BMTL | $50.4 \pm 0.1$ | $59.5 \pm 0.1$ | $65.6 \pm 0.1$ | $89.5 \pm 0.3$ | $92.3 \pm 0.2$ | $93.1 \pm 0.1$ | $65.7 \pm 0.4$ | $72.0 \pm 0.3$ | $76.8 \pm 0.3$ |
| VBMTL | $51.3 \pm 0.1$ | $60.9 \pm 0.1$ | $67.0 \pm 0.2$ | $90.8 \pm 0.6$ | $93.2 \pm 0.2$ | $93.5 \pm 0.1$ | $67.1 \pm 0.3$ | $73.0 \pm 0.7$ | $78.0 \pm 0.2$ |
| MTL-Uncertainty[1] | $51.8 \pm 0.1$ | $57.2 \pm 0.2$ | $66.8 \pm 0.2$ | $91.2 \pm 0.3$ | $93.8 \pm 0.2$ | $94.7 \pm 0.3$ | $74.6 \pm 0.2$ | $76.9 \pm 0.3$ | $79.2 \pm 0.3$ |
| MRN [9] | $57.4 \pm 0.1$ | $63.4 \pm 0.2$ | $67.1 \pm 0.1$ | $93.4 \pm 0.2$ | $94.8 \pm 0.3$ | $95.1 \pm 0.1$ | $73.7 \pm 0.4$ | $75.8 \pm 0.2$ | $79.7 \pm 0.3$ |
| LearnToBranch [55] | $38.3 \pm 0.5$ | $51.5 \pm 0.3$ | $62.2 \pm 0.4$ | $74.6 \pm 0.9$ | $80.4 \pm 1.2$ | $89.9 \pm 0.8$ | $51.7 \pm 0.9$ | $62.6 \pm 0.8$ | $71.6 \pm 0.4$ |
| TCGBML[59] | $52.8 \pm 0.1$ | $60.0 \pm 0.2$ | $68.7 \pm 0.2$ | $91.8 \pm 0.1$ | $95.0 \pm 0.2$ | $95.1 \pm 0.1$ | $73.9 \pm 0.3$ | $76.5 \pm 0.4$ | $79.3 \pm 0.4$ |
| MTVIB [60] | $49.9 \pm 0.2$ | $55.3 \pm 0.1$ | $66.2 \pm 0.1$ | $91.1 \pm 0.3$ | $94.1 \pm 0.3$ | $95.0 \pm 0.2$ | $74.0 \pm 0.4$ | $77.3 \pm 0.3$ | $78.9 \pm 0.5$ |
| VMTL [8] | $58.3 \pm 0.1$ | $65.0 \pm 0.0$ | $69.2 \pm 0.2$ | $93.8 \pm 0.1$ | $95.3 \pm 0.0$ | $95.2 \pm 0.1$ | $76.2 \pm 0.3$ | $77.9 \pm 0.2$ | $80.2 \pm 0.1$ |
| HNPs | $\mathbf{60.0} \pm 0.1$ | $\mathbf{66.2} \pm 0.2$ | $\mathbf{70.9} \pm 0.2$ | $\mathbf{94.6} \pm 0.1$ | $\mathbf{95.4} \pm 0.1$ | $\mathbf{95.8} \pm 0.1$ | $\mathbf{76.4} \pm 0.1$ | $\mathbf{79.5} \pm 0.1$ | $\mathbf{80.9} \pm 0.1$ |

**Results and Discussions.** We provide more comprehensive comparisons on `Office-Home`, `Office-Caltech` and `ImageCLEF` in Table 8. The best results are marked in bold. Our HNPs achieve competitive and even better performance on conventional multi-task classification datasets with different train-test splits. VSTL and VBMTL perform better than STL and BMTL, demonstrating the benefits of Bayes frameworks. Compared with multi-task probabilistic baselines, including VBMTL, TCGBML, MTVIB and VMTL, our HNPs can model more complex functional distributions with more powerful priors by inferring both representations and parameters for predictive functions.

Compared with VMTL [8], which neglects the hierarchical architecture of latent variables, the proposed HNPs show better performance. This demonstrates that by modeling the complex dependencies between heterogeneous context sets within the hierarchical Bayes framework, HNPs explore task-relatedness better. The hierarchical Bayes framework enables our model to explore the relevant knowledge even in the presence of distribution shifts among tasks.

## H  Application to Brain Image Segmentation

To show that HNPs have the potential to be helpful in settings other than categorization and regression, we apply the proposed HNPs to brain image segmentation.

**Dataset and Settings.** We adopt a brain image dataset [61] with lower-grade gliomas collected from 110 patients. The number of images varies among patients from 20 to 88. The goal is to segment the tumor in each brain image by predicting its contour.

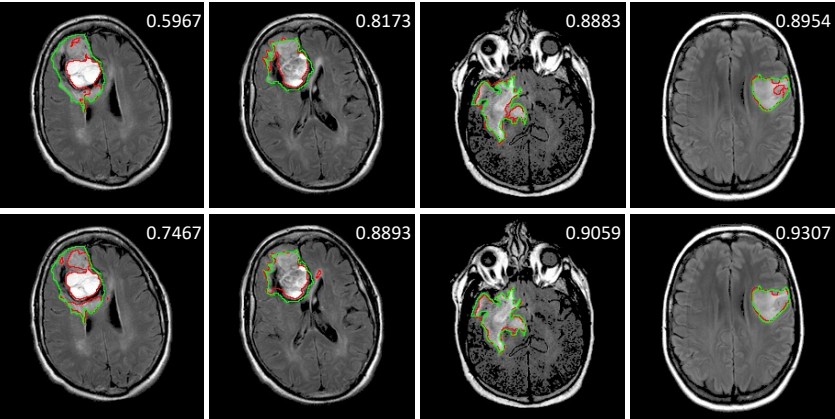

Figure 2: **Segmentation results by HNPs (bottom row) and U-Net (upper row)**. The ground truth contours are in green, and the predicted ones are in red. The numbers are DSC scores computed against the ground truth. HNPs can predict contours closer to the ground truth ones, indicating the advantages of exploring spatial context information for image segmentation.

We reformulate the segmentation task as a pixel-wise regression problem, where each pixel corresponds to a regression task to predict the probability of this pixel belonging to the tumor. In doing so, the spatial correlation and dependency among pixels are effectively modeled by capturing task-relatedness. For the task $m$, we define $\Omega_m$ as a local region centered at the spatial position $m$, which provides the local context information. In this case, the region centered at the pixel provides the local context information. Each task incorporates the knowledge provided by related tasks into the context of the predictive function. This offers an effective way to model the long-range interdependence of pixels in one image. In the implementation, we use U-Net [62] as the backbone and append our model to the last layer.

**Results and Discussions.** The proposed HNPs surpass the baseline U-Net by 0.8% (91.2% v.s. 90.6%) in terms of dice similarity coefficients (DSC) for the overall validation set. We provide the predicted contours by HNPs (bottom row) and the U-Net (upper row) in Figure 2, where the green outline corresponds to the ground truth and the red to the segmentation output. This figure shows that

HNPs predict contours closer to the ground truth. The results demonstrate the advantages of HNPs in exploring spatial-relatedness for medical segmentation.