# OpenReview forum: "Episodic Multi-Task Learning with Heterogeneous Neural Processes"
_NeurIPS.cc/2023/Conference — NeurIPS 2023 spotlight_

### Official Review · Reviewer_DuFB · 2023-07-03

**Soundness:** 4 excellent
**Presentation:** 3 good
**Contribution:** 4 excellent
**Rating:** 8
**Confidence:** 4

**Summary:**

The article is innovative in addressing the problem of data insufficiency by introducing the mutli-task learning method into the meta-learning paradigm. In this process, the author identify the issue of task-relatedness between heterogeneous tasks in a single episode and propose a solution called Heterogeneous Neural Processes (HNPs) for the episodic multi-task set-up.

The strengths of the article include:
1) The author clearly explains the differences and connections between the proposed method and existing works, particularly the contribution of handling heterogeneous tasks and distribution shifts.
2) The article provides impressive theoretical proofs and derivation processes of the algorithm in the appendix.
3) The code of the method is provided in the appendix, ensuring reproducibility.
4) The ablation study is comprehensive in exploring the impact of different factors on the method's performance, which is also impressive.

The weaknesses of the article:
The number of datasets used in the experiments is limited. While, the article provides more experiments in the appendix to supplement this issue. It does not significantly affect the credibility of the article's results. It would be helpful to provide more experimental results for the classification and regression tasks if possible.

**Strengths:**

As already written in summary, this paper is impressive considering its innovation:

1) The author clearly explains the differences and connections between the proposed method and existing works, particularly the contribution of handling heterogeneous tasks and distribution shifts.
2) The article provides impressive theoretical proofs and derivation processes of the algorithm in the appendix.
3) The code of the method is provided in the appendix, ensuring reproducibility.
4) The ablation study is comprehensive in exploring the impact of different factors on the method's performance, which is also impressive.

**Weaknesses:**

The number of datasets used in the experiments is limited. While, the article provides more experiments in the appendix to supplement this issue. It does not significantly affect the credibility of the article's results. It would be helpful to provide more experimental results for the classification and regression tasks if possible.

**Questions:**

I wonder if any more experiments are conducted on other datasets, including classification and regression tasks. How is the performance of the proposed method? Especially considering the limitations that the author mentions in the Conclusion.

**Limitations:**

The authors adequately addressed the limitations in the Conclusion part.

---

> ### Author Rebuttal · Authors · 2023-08-08
>
> *We appreciate Reviewer DuFB for the positive review and helpful feedback. We are glad the review found that our paper is innovative and impressive.*
>
> ---
>
> **1. More experimental results on new benchmarks or datasets.**
>
> Thanks for this suggestion. We’ve performed new experiments on the Office31 dataset [c, d] under the "3task5way1shot" setup. The dataset contains 31 object categories in three domains: Amazon, DSLR, and Webcam.
>
> |            Methods |      ERM      |   Proto.Net   |      CNPs     |      NPs      |     TNP_D     |      Ours     |
> |-------------------:|:-------------:|:-------------:|:-------------:|:-------------:|:-------------:|:-------------:|
> | Average   Accuracy | 63.53   ±0.71 | 64.54   ±0.64 | 49.02   ±0.74 | 40.52   ±0.75 | 69.69   ±0.87 | 71.89   ±0.52 |
>
> From this table, we find that our method outperforms baseline methods, demonstrating our model's effectiveness in addressing the data insufficiency under the episodic setup. We will add this table and related discussions in the final version of the paper.
>
> **2. Experiments with considering the limitation in the conclusion section.**
>
> Experiments in Appendix F avoid the negative influence of relabeling scheme by setting the conventional multi-task learning without the episodic training scheme. The related results are shown in Appendix F, where we apply the Office-Caltech and ImageCLEF datasets for classification and the Rotated MNIST dataset for regression. Empirical observations show that HNPs consistently outperform baselines, suggesting the effectiveness of the proposed model.
>
> **References:**
>
> [[c] K. Saenko, B. Kulis, M. Fritz and T. Darrell. "Adapting Visual Category Models to New Domains." ECCV2010.]( https://link.springer.com/chapter/10.1007/978-3-642-15561-1_16)
>
> [[d] B. Kulis, K. Saenko, and T. Darrell. "What You Saw is Not What You Get: Domain Adaptation Using Asymmetric Kernel Transforms." CVPR2011.]( https://ieeexplore.ieee.org/document/5995702)
>
> ---
>
> *We will add these collected results to the final version of the paper. We hope this rebuttal solidifies your positive outlook on our work, and we are happy to discuss it if you need further clarification. Thank you!*

---

> > ### Comment · Reviewer_DuFB · 2023-08-20
> > **Reply to author rebuttal**
> >
> > The experimental result shows efficacy. Good work. I have no further questions.

---

### Official Review · Reviewer_xREG · 2023-07-06

**Soundness:** 3 good
**Presentation:** 3 good
**Contribution:** 3 good
**Rating:** 6
**Confidence:** 2

**Summary:**

This paper proposes HNP: Heterogeneous Neural Processes, an approach that combines multitask learning with metalearning to address the problem of insufficiency of data in the multitask learning setting. The paper introduces episodic multitask learning as a way to exploit task relatedness from meta-training to meta-test episodes. HNP combines a global latent representation and a set of local parameters to model each task-specific function, with each local latent-parameter conditioned on the global representation. The inference module for prior and posterior variational distributions is implemented through a transformer-structure where meta-knowledge is instantiated as learnable tokens. Experimental results are provided for both regression and classification tasks, in comparison to other NP-based, multitask learning and metalearning methods.

---Rebuttal---
I read the authors rebuttal and considered it, along with other reviews. I left my score unchanged since I think this already reflected my overall evaluation of the paper.

**Strengths:**

- Regarding originality, the paper makes what seems to be a relevant contribution to the field in the sense of proposing a method that combines the benefits of metalearning with multitask learning for contexts where tasks are related, relying on a neural process-based approach.

- Regarding significance, the paper shows important improvements mainly in classification tasks, compared to baselines in NP-based models, multitask-based models and metalearning-based approaches.

- The quality and clarity of the paper are remarkable. The paper is very well-written and easy to follow, even for people not specifically into NP-based approaches.

**Weaknesses:**

- The results on regression tasks (Figure 4) seem to show that approaches such as ANPs and even NPs only can be quite competitive with respect to the proposed HNPs for at least a few of the tasks. Results in classification tasks, however, seem to denote much higher advantages of HNPs, especially in 1-shot learning, and considering the much larger number of tasks (or classes, in this case). This leads me to guess that the number of tasks has an influence on the amount, and quality, of sharing knowledge for HNPs. Does this imply that the method relies on the number of tasks to be large? What is the influence of the number of tasks, and the degree of relatedness of these tasks, in the way in which the local parameters and the global latent representation interact with each other, if any?

**Questions:**

- Please refer to the "weaknesses" section.

**Limitations:**

Limitations have been addresses. There are no potentially negative societal impact of this work.

---

> ### Author Rebuttal · Authors · 2023-08-08
>
> *We thank Reviewer xREG for the positive review and constructive feedback. We are glad that the review found that the quality and clarity of our paper are remarkable; the paper is very well-written and easy to follow.*
>
> ---
>
> **1. Influence of the number of tasks.**
>
> Thanks for the insightful questions. We’ve performed experiments and investigated the performance by gradually increasing tasks during inference.
>
> | Number   of tasks  |        1        |       2      |        3       |        4       |
> |--------------------:|:---------------:|:------------:|:--------------:|:--------------:|
> | Average   accuracy | 64.33     ±0.85 | 70.75 ±0.67 | 74.95    ±0.57 | 76.29    ±0.51 |
>
> From this table, we observe the average accuracy increases with more tasks. The main reason is that more tasks can provide richer transferrable information. Our model benefits from the positive transfer among tasks, and thus can obtain higher performance gain from more tasks.
>
>
> **2. Influence of the degree of task-relatedness.**
>
> The degree of task-relatedness is particularly essential to the learning of hierarchical latent variables. Global latent representation is task-specific and controls the access of each local latent parameter to all context sets in the episode for the corresponding task. With a higher degree of task-relatedness, such as fewer concept shifts among tasks, each local latent parameter can access richer prediction-aware information from related tasks, yielding a more discriminative classifier in practice.
>
> ---
>
> *We hope this rebuttal solidifies your positive outlook on our work, and we are happy to discuss it if you need further clarification. Thank you.*

---

> > ### Comment · Reviewer_xREG · 2023-08-19
> > **Response to authors**
> >
> > Thanks to the authors for their responses to my questions. I am happy with the numerical experiments in response to my question about the number of tasks. Regarding the degree of task-relatedness, are there any experimental results that you can provide to support your claims?

---

> > > ### Author Response · Authors · 2023-08-20
> > >
> > > *Thank you for your thoughtful review and positive remarks on our responses. We're pleased that the numerical experiments regarding the number of tasks were satisfactory to you.*
> > >
> > > ---
> > >
> > > Regarding the degree of task-relatedness, it is challenging to quantify the degree of task-relatedness theoretically in the literature. Instead, we found some intuitive observations to compare the degree of task-relatedness. For the Office-home dataset, there are four different tasks: “Art”, “Clipart”, “Painting” and “Real World”. As we all know, “Product” and “Real_world” are all collected from natural environments with rich expressiveness, but “Clipart” is artificial and simplified. In practice, we consider that the degree of task-relatedness between “Product” and “Real_world” is higher than that between “Real_world” and “Clipart”.
> > >
> > > To show the influence of the degree of task-relatedness, we’ve performed experiments under the 5way1shot setting on Office-home and investigated the performance by changing combinations of tasks during inference.  The following table shows that “Product + Real_world” performs 3.91% better than “Product + Clipart”. This demonstrates that “Product” can obtain higher performance gain from its more relevant “Real_world” than “Clipart”. These experimental results are consistent with our previous observation and support that higher task-relatedness can yield a more discriminative classifier.
> > >
> > > |  Types of combination                                 |   Product   | Product+Art | Product+Clipart | Product+Real_world |
> > > |-----------------------------------|:-----------:|:-----------:|:---------------:|:------------------:|
> > > |                    Acc. of Product | 75.92 ±0.88 | 78.97 ±0.78 |   77.49 ±0.84   |     81.40 ±0.75    |
> > > | Increased Acc. of Product|      0      |    +3.05    |      +1.57      |        +5.48       |
> > >
> > > ---
> > >
> > > *Your feedback is immensely valuable, and we'll ensure any new findings are included in the final version of the paper. Thank you!*

---

### Official Review · Reviewer_LXqe · 2023-07-07

**Soundness:** 4 excellent
**Presentation:** 4 excellent
**Contribution:** 4 excellent
**Rating:** 7
**Confidence:** 2

**Summary:**

The paper tackles the problem of learning tasks with insufficient data within the context of multi-task meta-learning. Specifically, the method proposed, namely Heterogenous Neural Processes (HNPs), is designed to leverage information amongst tasks in a single episode to improve performance while retaining learning from previous episodes during episodic training. The model leverages the function distributions task-wise prior to modeling each task in the episode while using a global prior and learned weights to meta-learn from previous episodes. A transformer architecture is adopted for inference in HNPs. Experiments on regression and classification tasks demonstrate the empirical improvements of HNPs over NP baselines and other methods.

**Strengths:**

- HNPs are interesting and the architectural choices in the model are well-motivated and supported with ablation studies.
- Experiments clearly establish the efficacy of HNPs in regression and classification tasks.
- Paper is well-written and uses proper notation which helps with understanding the submission better.

**Weaknesses:**

- Multi-task meta-learning is an interesting problem paradigm that may not possess the same applicability as meta-learning and multi-task learning do in applied settings individually. What are some examples of real-life environments where multiple tasks are needed to process during a single episode beyond the benchmarks experimented on?
- Classification performance is compared to older baselines, especially in the context of meta-learning methods. More recent baselines should be included for completeness of comparison.

**Questions:**

See the section above.

**Limitations:**

There is a clear discussion of the potential limitations of HNPs, although there is no explicit discussion of potential negative impacts; which I strongly encourage the authors to include in the submission.

---

> ### Author Rebuttal · Authors · 2023-08-08
>
> *We thank Reviewer LXqe for the positive review and helpful comments. We are glad that the review found that HNPs are interesting, the architectures are well-motivated, and the paper is well-written.*
>
> ---
>
> **1. Real-world examples or benchmarks in multi-task meta-learning.**
>
> Episodic multi-task learning has several potential applications in the real world, such as autonomous driving and robotic manipulations. In detail, the autonomous driving system needs to deal with different and related sensor data in an environment. However, the driving environment constantly changes along with the weather, time, country, etc. Thus, fast adapting the current multi-tasker to new environments is challenging in this industry track and our method can be a plausible solution for this challenge.
>
> **2. More experimental comparisons.**
>
> Thanks for your suggestion. We’ve included a recent multi-task and meta-learning method, such as MTL-bridge [a] and MLTI [b], and reported results on the Office-home dataset under the 4way5way1shot and 4way5way5shot setups as follows:
>
> | Methods      |  1-shot      | 5-shot      |
> |--------------|--------------|-------------|
> | MTL-bridge [a]| 64.31 ± 0.55 | 75.10 ±0.51 |
> | MLTI [b]      | 70.69 ± 0.73 | 79.59 ±0.58 |
> | Ours         | 76.29 ± 0.51 | 80.80 ±0.42 |
>
> From this table, we find that our method significantly outperforms other baselines with severely insufficient data, such as 1-shot. This is consistent with the conclusion obtained in the main paper. We will add these results and discussion in Table 1 of the final version.
>
> **References:**
>
> [[a] Wang, Haoxiang, Han Zhao, and Bo Li. "Bridging multi-task learning and meta-learning: Towards efficient training and effective adaptation." ICML2021.](https://arxiv.org/abs/2106.09017)
>
> [[b] Yao, Huaxiu, Linjun Zhang, and Chelsea Finn. "Meta-learning with fewer tasks through task interpolation." ICLR2022.](https://arxiv.org/abs/2106.02695)
>
> ---
>
> *We will add these collected results to the final version of the paper. We hope this rebuttal solidifies your positive outlook on our work, and we are happy to discuss it if you need further clarification. Thank you!*

---

> > ### Comment · Reviewer_LXqe · 2023-08-16
> >
> > The authors have adequately addressed the limitations noted in my original review. After further reviewing the rebuttal, and discussions by other reviewers, I will maintain my initial recommendation for acceptance of the submission.

---

### Official Review · Reviewer_21Dc · 2023-07-11

**Soundness:** 2 fair
**Presentation:** 3 good
**Contribution:** 2 fair
**Rating:** 6
**Confidence:** 3

**Summary:**

Learning multiple tasks with limited data poses a significant challenge. Addressing this issue, the paper proposes a novel approach that leverages heterogeneous information across tasks by combining the benefits of multi-task learning and meta-learning. This is achieved through the introduction of hierarchical neural processes (HNPs) within a hierarchical Bayes framework. To approximate the intractable integral in the posterior, variational inference is employed, facilitated by a transformer inference module. The performance of the proposed HNPs is thoroughly evaluated on both regression and classification tasks. The results demonstrate improved performance compared to typical baselines, highlighting the efficacy of the approach. The integration of multi-task learning and meta-learning through the HNPs framework showcases its potential for addressing the challenges associated with learning from limited data across multiple tasks.

**Strengths:**

* The concept of leveraging meta-learning and multi-task learning to enhance neural processes is both intriguing and well-grounded. The development of hierarchical neural processes (HNPs) is supported by established theoretical frameworks such as existing neural processes, hierarchical bases, and the natural alignment with variational inference.
* The incorporation of local and global latent representations to capture and share task-specific information is a logical choice, aligning well with the Bayesian interpretation of meta-learning. To facilitate approximate inference, a transformer-based inference module is employed, allowing for the fusion of meta-knowledge and information from various tasks.
* The experimental results showcase the adaptability of HNPs in both regression and classification tasks. Additionally, the conducted ablation studies shed light on the specific components of the models that influence the overall model performance, providing valuable insights into the model's behavior.

**Weaknesses:**

* While the utilization of neural processes in the proposed framework is intriguing, it is worth noting that the concept of combining multi-task learning and meta-learning is not novel. To provide a comprehensive understanding, it would be valuable for the authors to explicitly highlight the differentiating factors of their proposed work compared to existing approaches, as referenced in [1]-[3]. Furthermore, including these existing methods in the experimental comparisons would enhance the thoroughness and relevance of the evaluation. This would enable a more comprehensive analysis and facilitate a clearer understanding of the unique contributions and advantages of the proposed framework.
* The proposed framework imposes a constraint that the different tasks must share the same target space. This requirement significantly restricts the applicability of the method and narrows its scope. From the reviewers point of view, the framework aligns more closely with the domains of domain adaptation or multimodal learning rather than traditional multi-task learning. Acknowledging and discussing this limitation would provide a clearer understanding of the proposed method's potential use cases and its relationship with related research areas.
* References:
  * [1] “Multi-Task Meta Learning: learn how to adapt to unseen tasks”
  * [2] “Bridging Multi-Task Learning and Meta-Learning: Towards Efficient Training and Effective Adaptation”
  * [3] “Efficient and Effective Multi-task Grouping via Meta Learning on Task Combinations”

**Questions:**

* Why have “Multi-Task Processes” not been included in baselines? As the authors have
mentioned, it is highly relevant to this work.
*  Authors have claimed how the local and global representations capture task-specific functions but have not properly supported this claim experimentally other than showing performance gain by including the representations.
* Line 260 states that "... reliable uncertainty estimation ...", can the authors be more specific about "reliable"?
* The datasets used in classification tasks contain thousands of images, the reviewer was wondering how HNPs exactly handle the data-insufficiency issue.
* How about the time complexity of HNPs? Can the authors report the running time for example?


**Limitations:**

The authors have briefly discussed the limitation in Section 5. As mentioned in the Weakness session, the requirement of shared target space can be one potential limitation of the proposed method.

---

> ### Author Rebuttal · Authors · 2023-08-08
>
> *We appreciate Reviewer 21Dc the positive review and helpful feedback. We are glad that the review found that leveraging meta-learning and multi-task learning is intriguing and well-grounded.*
>
> ---
>
>
> **1. Related works [1-3] and MTPs.**
>
> Thanks for this suggestion.
>
> We highlight the differences between these related works and our work as follows:
>
> - [1] and MTPs are proposed for the single-input multi-output setting of multi-task learning in every single episode, where several tasks share the same input. However, in our work, episodic multi-task learning is designed based on the multi-input multi-output setting, where each episode suffers from distribution shifts between heterogeneous tasks, and each task has its individual input.
>
> - [2] theoretically addresses the conclusion that MTL methods are powerful and efficient alternatives to GBML for meta-learning applications. However, our method inherits the advantages of multi-task learning and meta-learning: simultaneously generalizing meta-knowledge from past to new episodes and exploiting task-relatedness across heterogeneous tasks in every single episode. Thus, our method is more suitable for solving the data-insufficiency problem.
>
> - [3] casts multi-task grouping as a meta-learning problem and aims to generalize the grouping ability from meta-train with a small number of combinations to meta-test with all the rest combinations. In [3], meta-train and meta-test share a fixed set of tasks. However, our work introduces episodic training to multi-task learning, where meta-test tasks differ from meta-train tasks. Thus, our method is more suitable to fast adapt the ability of learning task-relatedness to unseen tasks.
>
> We also provide experimental results of [2] on Office-home under the 4task5wayXshot setup.  Our method significantly outperforms [2] with severely insufficient data, such as 1-shot.
>
> | Methods      |  1-shot      | 5-shot      |
> |--------------|--------------|-------------|
> | MTL-bridge [2]  | 64.31 ± 0.55 | 75.10 ±0.51 |
> | Ours         | 76.29 ± 0.51 | 80.80 ±0.42 |
>
> Due to differences in setups between [1,3] MTP and our method, it is intractable to find standard implementations under our episodic multi-task setup. However, these interesting papers support our work that it is reasonable to meta-learn the ability of exploring task-relatedness by combining meta-learning and multi-task learning. We’ll cite these related papers and add discussions in the Line198 Related Work.
>
> **2. The same target space shared across different tasks.**
>
> This constraint is indeed the limitation of our work, and HNPs work under certain conditions. The shared target space is a simple way to ensure positive transfer among related tasks. However, the shared target space could not be the only condition to yield task-relatedness. If tasks have higher-level semantic relatedness, our work could be extended to the new case without the shared target spaces. But this requires more modifications to encode the target space.  We’ll add this discussion to the limitation section in Line 374.
>
> **3. More discussions on the local and global representations.**
>
> - In a theoretical sense, a hierarchical Bayesian framework with global and local latent variables yields a more rich and complex latent space to mitigate the expressiveness bottleneck, thus better parameterizing task-specific functions in stochastic processes.
>
> - In an empirical sense, global and local latent variables capture epistemic uncertainty in representation and parameter levels, respectively, and show improved performance in Table 2 and Figure 5 of the main paper.
>
> **4. Reliable uncertainty estimation in the regression of Line 260.**
>
> To show the reliability of our model, we’ve reported the negative loglikelihood (NLL) results in uncertainty quantification of regression tasks. The lower, the better.
>
> | Methods  | CNPs   | NPs    | ANPs    | Ours    |
> |----------|--------|--------|---------|---------|
> | Avg. NLL | 0.0935 | 0.8649 | -0.1165 | -0.5207 |
>
> The table shows that the proposed method obtains a lower average NLL, quantitatively outperforming other baselines in the toy-example experiment. We’ll add this discussion to the main paper.
>
> **5. Handling the data insufficiency issue.**
>
> For episodic multi-task classification, we feed the data in a meta-learning manner where each task suffers from the data-insufficiency problem. In detail, we define O-way K-shot as the context set of a single task, which first randomly samples O categories from the given label space and randomly samples K images for each individual category.
>
> To address this data insufficiency problem of the task, we propose to fully utilize the positive transfer from its related tasks in the same episode and reuse experiences from earlier episodes. Experimental results show that our model combining the advantages of both multi-task learning and meta-learning alleviates the data-insufficiency problem.
>
>
> **6. Runtime cost in the inference of models.**
>
> Thanks for this suggestion. The time complexity positively correlates with the number of Monte Carlo samples of latent variables. As shown in the right part of Figure 5, the runtime per iteration grows rapidly as the number of samples increases. For interests, we also compare the runtime cost of other NP-based methods with that of our model as follows:
>
> | Methods        | CNPs | NPs  | TNPs | Ours  |
> |----------------|------|------|------|-------|
> | inference time(s) | 0.04 | 0.05 | 0.08 | 0.15  |
>
> This table shows our model needs a bit more inference time than other NP-based methods for performance gains. The cost mainly comes from inferring the designed hierarchical latent variables, however, we consider this a worthwhile trade-off for the extra performance. We will add this limitation to our conclusion section.
>
> ---
>
> *We hope this rebuttal solidifies your positive outlook on our work, and we are happy to discuss it if you need further clarification. Thank you.*

---

### Official Review · Reviewer_YPdL · 2023-07-21

**Soundness:** 4 excellent
**Presentation:** 3 good
**Contribution:** 3 good
**Rating:** 6
**Confidence:** 4

**Summary:**

The paper proposes an NP-based approach to episodic multi-task learning called HNPs that uses transformer-based hierarchically arranged inference modules to model class-specific distributions. As a result, the predictions are obtained by a dot product of the input features and the sampled class-specific latent variables, thus making the latent variables prediction-aware.

**Strengths:**

- The paper focuses on the setting of episodic multi-task learning and proposes Heterogenous Neural Processes (HNPs) -- a new member to the NP family for classification and regression tasks.
- The authors provide a well-motivated and well-explained method that emphasizes the potential of HNPs. A good highlight of their method remains a two-layer cross-task and intra-task arrangement of latent variables where the latter variables are prediction-aware.
- The authors adapt transformer encoder layers as inference modules for HNPs which is a good alternative to traditional MLP-based NP layers that lack ample inductive biases, particularly for classification tasks.
- The experiments and ablations look quite diverse ranging from episodic multi-task regression to classification settings.

**Weaknesses:**

- While the NP-based methods in Table 1 sound fairly recent, the methods used from multi-task learning and meta-learning branches remain fundamental. Can the authors look into comparing their method with more up-to-date methods from these two branches?
- The main table of results compares HNPs with a range of other NPs with and without transformer-based inference modules. Besides Figure 5, it would also be valuable for the authors to provide a comparison of the computational and/or runtime cost at inference for these models.
- It would be valuable to clarify further the working of the inference modules under a more generic classification setting. E.g., does the episodic multi-task learning setup guarantee that each training batch consists of samples from all possible classes? In particular, if a minibatch has samples from a few classes missing, how does the HNP handle this given that the prediction-aware transformer encoder layer (possibly) expects samples from classes to be arranged as a sequence?
- Can the authors comment on the saturating gap between the performance of HNPs and TNP-D in settings with a larger number of classes? It would be interesting to see an analysis of the performance comparison between the two on settings with more number of classes.
- While the authors show that the probabilistic version of the HNP outperforms its deterministic counterpart (Table 3), it would be valuable to mention how well the probabilistic HNP makes use of the prior. In particular, how the two KL values in eq. (4) vary with training, do they approach to nearly zero in the latter training stages? And if so, is the model carefully considering the prior?



**Questions:**

- Further on the use of priors, NPs are known to suffer from posterior collapse, and in classification settings, it could be the case that the main classification loss (cross-entropy, etc.) dominates over the posterior approximation loss (KL div) to the extent where the model could perform equally well without the latter loss. A comparison with and without the KL losses could be useful here.
- A standard framework for NP uses a concatenation of inputs x with outputs y to be fed into the NP layers. However, the HNP does not seem to be using such concatenated features. Can the authors clarify if they have any observations with and without such concatenated features for generative modeling?
- Can the authors clarify further how they make use of the context set at train time vs test time?

**Limitations:**

Overall, this paper presents a valuable contribution to the field of episodic multi-task learning. However, the current empirical results may limit its impact and applicability in real-world scenarios. Therefore, the authors may want to elaborate further on their empirical evaluations,  include more recent methods for comparison, discuss their saturating performance with more number of classes, or alternatively, provide a clear explanation for this shortcoming.

---

> ### Author Rebuttal · Authors · 2023-08-08
>
> *We thank Reviewer YPdL for the positive review and constructive feedback. We are glad that the review found that our method is well-motivated and well-explained.*
>
> ---
>
> **1. More experimental comparison and discussions.**
>
> We’ve reimplemented recent multi-task and meta-learning methods, such as, MTL-bridge [a] and MLTI [b] on Office-home under the 4way5way1shot and 4way5way5shot setups as follows:
>
> | Methods      |  1-shot      | 5-shot      |
> |--------------:|:--------------:|:-------------:|
> | MTL-bridge [a]| 64.31 ± 0.55 | 75.10 ±0.51 |
> | MLTI [b]      | 70.69 ± 0.73 | 79.59 ±0.58 |
> | Ours         | 76.29 ± 0.51 | 80.80 ±0.42 |
>
> From this table, we find that our method significantly outperforms other baselines with severely insufficient data, such as 1-shot. This is consistent with the conclusion obtained in the main paper.
>
> **2. Runtime cost in the inference of models.**
>
> We’ll include comparisons on runtime cost per iteration of models as follows:
>
> | Methods        | CNPs | NPs  | TNPs | Ours |
> |----------------:|:------:|:------:|:------:|:-------:|
> | Inference time(s) | 0.04 | 0.05 | 0.08 | 0.15  |
>
> From this table, we can see our model needs a bit more inference time than other NP-based methods for performance gains. The cost mainly comes from inferring the designed hierarchical latent variables; however, we consider this a worthwhile trade-off for the extra performance. We will add this limitation to our conclusion section.
>
> **3. Each training batch consists of all possible classes?**
>
> Thanks for this question.
>
> Yes, we indeed include all possible classes in a single episode during the training and testing times. Under the episodic setup, e.g., O-way K-shot, the inference module works on the task level rather than the sample level. Thus, the module is fed with data from a whole task, where the context set contains samples from all O classes.
>
> Moreover, when samples from a few classes are missing, we can also provide an alternative solution to this scenario. As shown in the formulation of our model, latent variables are inferred from the corresponding samples and learnable tokens. Without the samples, the transformer encode layer only takes the learnable tokens as input and outputs the distribution of the corresponding latent variables.
>
> **4. Discussion on the gap between HNPs and TNP-D’s performance with a larger number of classes.**
>
> We’ve performed the experiment on the DomainNet dataset and reported the result as follows:
>
> |     Methods    |         5way       |        20 way      |        25way       |        30way       |        35way       |        40way       |
> |---------------:|:------------------:|:------------------:|:------------------:|:------------------:|:------------------:|:------------------:|
> |        TNPs    |     49.10 ±0.42    |     28.83 ±0.17    |     25.93 ±0.14    |     24.08 ±0.12    |     22.62 ±0.11    |     21.64 ±0.10    |
> |        HNPs |     62.36 ±0.53    |     39.32 ±0.23    |     35.72 ±0.19    |     32.27 ±0.17    |     31.27 ±0.14    |     29.31 ±0.13    |
>
> From this table, we find that our method consistently outperforms TNPs as the number of classes increases from 20 to 40 in step 5. The performance gap between them narrows slightly. The main reason could be that the setting with more classes suffers from less data insufficiency.
>
> **5. Role of probabilistic HNPs, KL values in meta training, and without KL scenarios.**
>
> - The probabilistic HNPs encode the context as the heterogeneous prior and reveal the uncertainty resulting from data insufficiency and the extent of observations in tasks. Additionally, minimizing KL terms encourages priors inferred from context sets to stay close to posteriors inferred from target sets, guiding more efficient conditional generation.
>
> - In training processes, we observed the KL divergence value does not decrease to 0 after convergence, e.g., KL values in the scale of e-1 on Office-home. This is also part of traits in NPs family, suggesting the approximate posterior and the approximate prior encode different conditional information during the generation of latent variables.
>
> - Note that our primary contribution is a new variant of the stochastic NP model. Without the constraint of the KL divergence, this no longer constitutes the stochastic NP model, and we cannot find the appropriate stochastic NP baseline to compare. Instead, we consider the CNP as the appropriate baseline without the KL divergence in the discussion and results are already reported in Table 1 of the main paper.
>
> **6. Concatenation of $x$ and $y$ and observations.**
>
> Thanks for this insightful question. For various benchmarks, there are different operations w.r.t. the couple of $x$ and $y$ in our model. In regression scenarios, we concatenate the x and y as the input of the inference layers, as done in vanilla NPs. In classification scenarios, we exploit the label information to infer the local latent variables by incorporating the prediction-aware token and context samples from the corresponding classes. In practice, this operation helps us to generate more discriminative local latent parameters.
>
> **7. Use of the context set at train vs. test time.**
>
> Thanks for this question. During the meta-test, we make predictions with priors inferred from the context set. This is the same as that in meta-training. Also, note that target sets are unseen to infer the corresponding posteriors in the meta-test.
>
> **References:**
>
> [[a] Wang, Haoxiang, Han Zhao, and Bo Li. "Bridging multi-task learning and meta-learning: Towards efficient training and effective adaptation." ICML2021.](https://arxiv.org/abs/2106.09017)
>
> [[b] Yao, Huaxiu, Linjun Zhang, and Chelsea Finn. "Meta-learning with fewer tasks through task interpolation." ICLR2022.](https://arxiv.org/abs/2106.02695)
>
> ---
>
> *We hope this rebuttal solidifies your positive outlook on our work, and we are happy to discuss it if you need further clarification. Thank you.*

---

> > ### Comment · Reviewer_YPdL · 2023-08-17
> >
> > Thank you for your comment. This clears out some of my concerns. I still have a few questions regarding the architectural choices of the proposed model:
> >
> > - Given that the class-specific probabilistic encoder is conditioned (partly) upon the outputs of the task-specific encoder, why in particular use a Transformer Encoder layer for the former again? Put another way, could a Transformer-styled Decoder layer instead be used for deriving the class-specific encodings? Why / why not?
> > - Can the authors show some empirical validation of the prior being used in their classification model? E.g., a graph of the KL divergence vs training progress.
> > - Can the authors comment on the setting on which the reported inference time costs have been computed? Is it classification or regression? Which dataset?

---

> > > ### Author Response · Authors · 2023-08-17
> > >
> > > *Thanks for your time and suggestions! We're glad to have a further discussion on these issues.*
> > >
> > > ---
> > >
> > > **1. Architectural choice.**
> > >
> > > Thank you for this question. Using Transformer Encoder layers in a unified manner for the local latent parameters and the global latent representations offers benefits due to their self-attention mechanism and the introduced meta-knowledge tokens. In detail, the task-specific transformer-structured inference module $\theta$ can capture intra-task relationships to enhance the task-specific token. Meanwhile, the class-specific transformer inference module $\phi$ can refine the class-specific tokens with discriminative information by exploring inter-task relationships.
> > >
> > > A Transformer-styled Decoder could also be used to infer the class-specific latent parameters. In this case, each Monte Carlo sample of task-specific latent representation is taken as an extra input token of a decoder layer. However, our previous ablation studies found that this way deteriorates the inference process (68.16 ±0.54 vs. 76.29 ±0.51 under the 4task5way1shot classification setting on the Office-home dataset). A possible reason can be that our inference approach keeps more task-specific information to mitigate the distribution shifts by directly merging each Monte Carlo sample of task-specific latent representation with the refined class-specific tokens. Thus, we chose the Transformer Encoder structure as our inference module in this paper.
> > >
> > > **2. Graph of the KL divergence vs. training progress.**
> > >
> > > Thank you for this suggestion. We've shared the trend of the KL divergence under the 4task5way1shot classification setting on the Office-home dataset in an anonymous link with the area chair according to the guideline. We hope the respected area chair will forward that to you. This figure shows that as the number of iterations increases, the KL divergence value decreases but does not reach 0, and convergences in the scale of e-1. We will add this visualization and discussions to our final version.
> > >
> > > **3. The setting of the reported inference time costs.**
> > >
> > > Thank you for this question. We've provided the inference time cost under the 4task5way1shot classification setting on the Office-home dataset.

---

### Official Review · Reviewer_n4KJ · 2023-07-26

**Soundness:** 3 good
**Presentation:** 3 good
**Contribution:** 2 fair
**Rating:** 6
**Confidence:** 4

**Summary:**

The paper introduces a meta-learning paradigm to perform few-shot multi-task learning. It introduces a local latent representation $\omega$, which effectively combines heterogeneous information from various tasks. Additionally, the paper suggests using transformer-structured inference modules to infer both hierarchical latents for task-relatedness and learnable tokens as meta-knowledge. The experimental results demonstrate that the proposed method achieves good performance in the few-shot multi-domain image classification task.

**Strengths:**

- the problem formulation and the proposed method is novel to the best of my knowledge.
- the proposed meta knowledge and task-specific latent representation are well motivated.
- the method is proved to be effective in the few shot multi-domain image classification task.

**Weaknesses:**

- Minor improvement in 5-shot setting: while the proposed method shows promising improvement on 1-shot setting across the two datasets. The performance gap diminishes quickly when scale to 5-shot setting (less than 2% improvement v.s. TNP-D). It is questionable whether the method can maintain the superiority when number of unseen samples is further increased.
- Comparison Fairness: In the classification experiment, the paper evaluates the effectiveness of meta-learning methods (e.g., MAML) by treating each task separately. A more fair baseline implementation would merge all the tasks into a single task and provide the same amount of meta-training/meta-testing data. For instance, in a 4-task 20-way 1-shot experiment setup, baseline meta-learning methods should have a 20-way 4-shot setup.
- Evaluation Metrics: The claim regarding the regression task and the smoother predictive curves with reliable uncertainty estimation from HNPs lacks quantitative support and comes across as vague. Furthermore, the experimental setup from this single example raises doubts about whether it was cherry-picked.
- Clarification: The caption of Figure 1 stating, "different colors denote different categories" is not clear. Instead, different shades should be associated with class categories, while different colors (yellow, blue, red, etc.) should correspond to various training examples.

**Questions:**

- more thorough comparison and experiment discussion as stated in the weakness section.
- can the proposed method work when target space across tasks differs from each other?
- What exactly does meta-knowledge $\omega = \omega_{1:O}$ learn?


**Limitations:**

One limitation that can be highlighted is that the proposed method requires the target space to be strictly the same across all tasks during meta-training. This requirement could potentially limit the applicability of the method in realistic scenarios where taxonomies may differ across tasks.

---

> ### Author Rebuttal · Authors · 2023-08-08
>
> *We thank Reviewer n4KJ for the positive review and thorough feedback. We are glad that the review found that our method is novel, the proposed meta-knowledge and latent representation are well motivated.*
>
> ---
>
>
> **1. Performance with more “shot”.**
>
> Thanks for this question. We’ve performed additional experiments on the Office-home dataset by increasing the number of context samples. The results and discussions are:
> | Methods |    1-shot   |    5-shot   |   10-shot   |   20-shot   |
> |--------:|:-----------:|:-----------:|:-----------:|:-----------:|
> |    TNPs | 65.49 ±0.53 | 78.94 ±0.43 | 80.81 ±0.32 | 81.12 ±0.68 |
> |    Ours | 76.29 ±0.51 | 80.80 ±0.42 | 81.28 ±0.38 | 81.56 ±0.36 |
>
> We observe that the proposed method has more advantage over TNPs with the context data points below ten shots. It seems that with shots greater than ten, both methods will reach a performance bottleneck in fast adaptation.
>
>
> **2. New evaluation in classification.**
>
> Thanks for this suggestion. We conducted experiments with the requested 20-way 4-shot setup, which needs to mix samples from all tasks in one episode.
>
> |       Methods    |     4task20way1shot    |       20way4shot     |
> |-----------------:|:----------------------:|:--------------------:|
> |          MAML    |      34.29   ±0.19     |     37.23   ±0.25    |
> |     Proto.Net    |      32.72   ±0.18     |     37.12   ±0.22    |
> |          Ours    |      51.82   ±0.23     |          -           |
>
> As shown in the table, we find that MAML and Proto.Net perform better under the 20-way 4-shot but cannot outperform our method. The main reason is that our method can better handle distribution shifts among tasks by exploring task-relatedness rather than simply mixing them together.
>
>
> **3. Uncertainty quantification in the regression.**
>
> We’ve reported the negative loglikelihood (NLL) results in uncertainty quantification of regression tasks:
>
> | Methods  | CNPs   | NPs    | ANPs    | Ours    |
> |----------|--------|--------|---------|---------|
> | Avg. NLL | 0.0935 | 0.8649 | -0.1165 | -0.5207 |
>
> As shown in the table, the proposed method obtains a lower average NLL, quantitatively outperforming other baselines in the toy-example experiment.
>
>
> **4. Refining Figure 1.**
>
> Thank you for this suggestion. Different colors (yellow, blue, red, etc.) represent different label spaces among episodes; the same color with different shades (adding different percentages of black) represents different categories in the same task. We’ll polish Figure 1 in the final version.
>
>
> **5. Other questions.**
>
> * When the target space differs from each other, can the proposed method work?
>
>     Thanks for your insightful question.  This constraint of the same target space is indeed the limitation of our work, and HNPs could work under certain conditions. Our model is to explore task-relatedness across tasks in a single episode and considers this as a new kind of meta-knowledge. The shared target space is a simple way to ensure positive transfer among related tasks.
>
>     However, the shared target space could not be the only condition to yield task-relatedness. If tasks have higher-level semantic relatedness, our work could be extended to the new case without the shared target spaces. But this requires more modifications to encode the target space, so we’ll add this discussion to the limitation section in Line 374 in the final version.
>
> * What does meta-knowledge $\omega=\omega_{1:O}$ learn?
>
>     Thank you for this question. The $\omega=\omega_{1:O}$ learns the shared knowledge and inductive biases across all tasks, and its distribution at a parameter level can capture epistemic uncertainty in practice.
>
> ---
>
> *We will add these collected results to the final version of the paper. We hope that this rebuttal solidifies your positive outlook on our work, and we are happy to discuss it if you need any further clarifications. Thank you.*

---

> > ### Comment · Reviewer_n4KJ · 2023-08-16
> >
> > The authors have addressed most of my concerns in the rebuttal. However, I'm not fully convinced the proposed method can be easily extended to practical scenarios where tasks have diverse target space. Therefore, I maintained my score as weak accept. I encourage the authors to include the additional experiment results in the final version.

---

> > > ### Author Response · Authors · 2023-08-16
> > >
> > > We appreciate your feedback and recognition of our efforts in addressing your concerns. Regarding the concern about extending our method to diverse target spaces in practical scenarios, we acknowledge the challenge. We will add this to the limitation section in the final version. Thank you for your valuable suggestions.

---

### Author Rebuttal · Authors · 2023-08-08

*We thank all reviewers for their insightful comments and supportive suggestions. These feedbacks are constructive in improving this paper’s quality. Here, we provide a global response to summarize the primary merits of this paper and concerns to address as follows.*

---

## **I. Merits of this Work**

We summarize the following supportive points from reviewers:

* The proposed method is novel, well-motivated, original, and innovative by Reviewers **n4KJ/ YPdL/ LXqe/ xREG/ DuFB**.

* The paper is well-written with clear contributions by Reviewers **LXqe/ xREG/ DuFB**.

* Experimental results show HNPs’ effectiveness with diverse tasks and ablation studies by Reviewers **n4KJ/ YPdL/ 21Dc/ LXqe/ xREG/ DuFB**.

* The theoretical proofs and derivations of algorithms are impressive by Reviewer **DuFB**.

---

## **II. Primary Concerns**

**1. Additional experimental comparisons and discussions by Reviewers n4KJ/ YPdL/ 21Dc/ LXqe.**

(1) We’ve performed experiments on Office-home under the 4task5wayXshot setup.

| Methods |    1-shot   |    5-shot   |   10-shot   |   20-shot   |
|:--------:|:-----------:|:-----------:|:-----------:|:-----------:|
|    TNPs | 65.49 ±0.53 | 78.94 ±0.43 | 80.81 ±0.32 | 81.12 ±0.68 |
|    Ours | 76.29 ±0.51 | 80.80 ±0.42 | 81.28 ±0.38 | 81.56 ±0.36 |

Please refer to Response to Reviewer **n4KJ** for more details and discussions.

(2) We’ve performed experiments on DomainNet under the 6taskXway1shot setup.

|     Methods    |         5way       |        20 way      |        25way       |        30way       |        35way       |        40way       |
|:---------------:|:------------------:|:------------------:|:------------------:|:------------------:|:------------------:|:------------------:|
|        TNPs    |     49.10 ±0.42    |     28.83 ±0.17    |     25.93 ±0.14    |     24.08 ±0.12    |     22.62 ±0.11    |     21.64 ±0.10    |
|        Ours    |     62.36 ±0.53    |     39.32 ±0.23    |     35.72 ±0.19    |     32.27 ±0.17    |     31.27 ±0.14    |     29.31 ±0.13    |

Please refer to Response to Reviewer **YPdL** for more details and discussions.

(3) We've compared meta-learning baselines and our method on DomainNet under the 20way4shot setup.

|       Methods    |     4task20way1shot    |       20way4shot     |
|-----------------:|:----------------------:|:--------------------:|
|          MAML    |      34.29   ±0.19     |     37.23   ±0.25    |
|     Proto.Net    |      32.72   ±0.18     |     37.12   ±0.22    |
|          Ours    |      51.82   ±0.23     |          -           |

Please refer to Response to Reviewer **n4KJ** for implementation and discussions.

(4) We’ve implemented two recent works [a] and [b] and performed experiments on Office-home under the 4task5way1shot and 4task5way5shot setups.

| Methods      |  1-shot      | 5-shot      |
|--------------|--------------|-------------|
| MTL-bridge [a]   | 64.31 ± 0.55 | 75.10 ±0.51 |
| MLTI [b]         | 70.69 ± 0.73 | 79.59 ±0.58 |
| Ours         | 76.29 ± 0.51 | 80.80 ±0.42 |

Please refer to Response to Reviewer **YPdL/ 21Dc/ LXqe** for more details and discussions.

(5) We've provided the runtime cost in the inference of NP-based models per iteration.

| Methods        | CNPs | NPs  | TNPs | Ours  |
|----------------|------|------|------|-------|
| inference time(s) | 0.04 | 0.05 | 0.08 | 0.15  |

Please refer to Response to Reviewer **YPdL and 21Dc** for more details and discussions.

(6) We’ve reported the negative loglikelihood (NLL) results in uncertainty quantification of regression tasks (the lower, the better).

| Methods  | CNPs   | NPs    | ANPs    | Ours |
|----------|--------|--------|---------|---------|
| Avg. NLL | 0.0935 | 0.8649 | -0.1165 | -0.5207 |

Please refer to Response to Reviewer **n4KJ and 21Dc** for more details and discussions.

&nbsp;

**2. More benchmark datasets by Reviewer DuFB.**

We've validated the performance of methods on the Office31 [c, d] dataset.

|            Methods |      ERM      |   Proto.Net   |      CNPs     |      NPs      |     TNP_D     |      Ours |
|-------------------:|:-------------:|:-------------:|:-------------:|:-------------:|:-------------:|:-------------:|
| Average   Accuracy | 63.53   ±0.71 | 64.54   ±0.64 | 49.02   ±0.74 | 40.52   ±0.75 | 69.69   ±0.87 | 71.89   ±0.52 |

Please refer to Response to Reviewer **DuFB** for details and discussions.

&nbsp;

**3. Updates on the limitation of the shared target space by Reviewers n4KJ and 21DC.**

It is indeed challenging to ensure all tasks share the same target space. In this case, more modifications are required to encode the target space when using our method. We'll add this discussion to the limitation section in Line 374.

&nbsp;

**4. Other questions and future updates in the manuscript.**

For detailed concerns and questions, we respectively reply to individual reviewers. The newly collected experimental results, together with discussions, will be included in the final version.

&nbsp;

**References:**

[[a] Wang, Haoxiang, Han Zhao, and Bo Li. "Bridging multi-task learning and meta-learning: Towards efficient training and effective adaptation." ICML2021.](https://arxiv.org/abs/2106.09017)

[[b] Yao, Huaxiu, Linjun Zhang, and Chelsea Finn. "Meta-learning with fewer tasks through task interpolation." ICLR2022.](https://arxiv.org/abs/2106.02695)

[[c] K. Saenko, B. Kulis, M. Fritz and T. Darrell. "Adapting Visual Category Models to New Domains." ECCV2010.]( https://link.springer.com/chapter/10.1007/978-3-642-15561-1_16)

[[d] B. Kulis, K. Saenko, and T. Darrell. "What You Saw is Not What You Get: Domain Adaptation Using Asymmetric Kernel Transforms." CVPR2011.]( https://ieeexplore.ieee.org/document/5995702)

---

*Thank all reviewers and area chairs again for collaborating with us to improve our manuscript.*

---

### Decision · Program_Chairs · 2023-09-21

**Decision:**

Accept (spotlight)

**Comment:**

This paper proposes a meta-learning method for multi-task learning. The proposed method, called heterogeneous neural processes, can capture relationships among heterogeneous tasks and use knowledge for tasks with a limited amount of data. The experimental results demonstrate the effectiveness of the proposed method in both classification and regression problems. This paper is well-written. The problem formulation and the proposed method are novel and interesting. The architectures of the proposed model are well-motivated, and their advantages are supported by their ablation study. The additional experiments in the rebuttals strength the paper.